# Universal Murray's law for optimised fluid transport in synthetic structures

Binghan Zhou [1], Qian Cheng [2], Zhuo Chen [1], Zesheng Chen[1], Dongfang Liang [2], Eric Anthony Munro[1], Guolin Yun [1], Yoshiki Kawai[3], Jinrui Chen[1], Tynee Bhowmick[1], Padmanathan Karthick Kannan[4], Luigi Giuseppe Occhipinti [1], Hidetoshi Matsumoto [3], Julian William Gardner[4], Bao-Lian Su [5,6] & Tawfique Hasan [1] ✉

Materials following Murray's law are of significant interest due to their unique porous structure and optimal mass transfer ability. However, it is challenging to construct such biomimetic hierarchical channels with perfectly cylindrical pores in synthetic systems following the existing theory. Achieving superior mass transport capacity revealed by Murray's law in nanostructured materials has thus far remained out of reach. We propose a Universal Murray's law applicable to a wide range of hierarchical structures, shapes and generalised transfer processes. We experimentally demonstrate optimal flow of various fluids in hierarchically planar and tubular graphene aerogel structures to validate the proposed law. By adjusting the macroscopic pores in such aerogel-based gas sensors, we also show a significantly improved sensor response dynamics. In this work, we provide a solid framework for designing synthetic Murray materials with arbitrarily shaped channels for superior mass transfer capabilities, with future implications in catalysis, sensing and energy applications.

The performance of materials is strongly influenced by their structures[1,2]. Hierarchically branched materials, with their multi-level and interconnected pores at various scales are a prime example[2–4]. The larger interconnected pores in such materials shorten the transport path and improve mass transfer efficiency, while the branching pores with increasing numbers but smaller size enhance the specific surface area and active reaction sites[5,6]. Significant recent efforts have been devoted to the theoretical design and construction of such unique porous structures for optimised performance[7–9]. Towards this end, inspired by biological networks such as leaf veins and vascular systems, Murray's law has been considered for the construction of porous materials[10–22]. This biomechanical theory stipulates the optimal hierarchical porous network for the most efficient mass transfer and is poised to lead an era of 'nanostructuring by design'[1,9]. Indeed, recent synthetic porous constructs attempting to implement Murray's law have been claimed to improve application performances involving mass transfer processes, such as catalysis[12–14], sensing[11,16], and energy storage[18,19].

However, the original Murray's law cannot be appropriately implemented to synthetic nanostructured materials. This is because it was experimentally derived from branching circular tubes of biological networks in living organisms which cannot be replicated due to the limitations in current synthesis techniques. Therefore, the above attempts to fabricate Murray materials differ significantly from the original Murray's law in terms of pore shapes and locations. Such mismatch in pore geometries between the original theory and synthetic Murray materials strongly reduces the promised optimum mass transfer. This also means that there has not yet been any convincing

[1]Cambridge Graphene Centre, University of Cambridge, Cambridge CB3 0FA, UK. [2]Department of Engineering, University of Cambridge, Cambridge CB2 1PZ, UK. [3]Department of Materials Science and Engineering, Tokyo Institute of Technology, Tokyo 152-8552, Japan. [4]School of Engineering, University of Warwick, Coventry CV4 7AL, UK. [5]Laboratory of Inorganic Materials Chemistry (CMI), University of Namur, B-5000 Namur, Belgium. [6]State Key Laboratory of Advanced Technology for Materials Synthesis and Processing, Wuhan University of Technology, Wuhan 430070, China. ✉e-mail: th270@cam.ac.uk

experimental verification of the superior mass transport in synthetic Murray materials. Therefore, to exploit the benefit of Murray materials, the original law needs to be generalised such that it can go beyond a specific pore architecture to accommodate various other geometries commonly achievable through traditional synthesis.

In this work, we propose a Universal Murray's law with a generalised pore structure and mass transfer process. We prove the wide applicability of our approach in hierarchical networks with non-circular cross-sections and planar structures. To validate our proposal, we construct planar and tubular hierarchical structures using uni-directionally and bidirectionally freeze-cast graphene oxide aerogels (GOA), and experimentally confirm optimal mass transfer performance for laminar flow using a range of fluids. We further show how simple structural optimisation guided by our proposed theory yields a significant dynamic performance improvement in GOA-based gas sensors. Our study lays a solid theoretical foundation of Murray's law in synthetic hierarchically porous materials, with a broad scope for applications involving mass transfer.

## Results

### The proofs of Murray's law and its derivations

The original Murray's law was experimentally derived to describe how blood vessel structures offer the most energy-efficient transport[23]. This theory has been extensively studied in the field of biomechanics over the last decades[24–26]. Indeed, numerous transport networks in living organisms, such as vascular systems in animals, tracheal tubes in insects, and plant veins, have all been found to obey Murray's law or its derivations[27–30]. Based on the commonly found shapes in biological networks, Murray's law has thus far been largely applied to cylindrical pores, branching pipe networks with circular channels; Fig. 1a. The original law was initially obtained by minimising the sum of the work required to overcome the flow resistance and the metabolic cost of vascular systems (for details, see Supplementary Note 1):

$$\sum r_1^3 = \sum r_2^3 = \sum r_3^3 = \cdots \tag{1}$$

This equation proposes that the hierarchical pipe network becomes optimal for laminar flow when the sum of the cubes of the tube radii at each branching level is constant; Fig. 1a. The furcating point in the Murray network also shows that the radius cube of the parent channel is equal to the sum of the children channels'. Beyond laminar fluid flow, Murray's law takes on a square form for diffusion, ionic migration in electrolyte under an applied electric field, and electron transportation in conductors or semiconductors[27]:

$$\sum r_1^2 = \sum r_2^2 = \sum r_3^2 = \cdots \tag{2}$$

Later, it was proposed that vascular systems can be optimised within the confines of a given total volume instead of considering tissue metabolism by following various forms of Murray's law described earlier[27,31,32]. This alternative approach enables the application of Murray's law in optimising the mass transfer of synthetic porous materials, as the 'mass transfer performance' of materials should be considered as an intensive property, referring to the mass transport capacity per unit volume. The optimisation of porous materials under the volume constraint essentially corresponds to the optimal mass transfer performance.

Additionally, the transported mass is assumed to remain conserved in the initial Murray's law for biological networks. Considering the reaction or adsorption of materials in hierarchical pores, the generalised Murray's law[1,10] introduced a mass loss ratio $X$:

$$r_0^\alpha = \frac{1}{(1-X)} \sum r_i^\alpha \tag{3}$$

where $r_0$ represents the radius of the parent channel and $r_i$ for the branched children channels. The exponent $\alpha = 3$ for laminar flow and $\alpha = 2$ for diffusion or ionic migration. The surface area of pores at different levels can be adeptly utilized to derive $X$. This mass loss ratio $X$ can also be directly substituted into all subsequent results in this work.

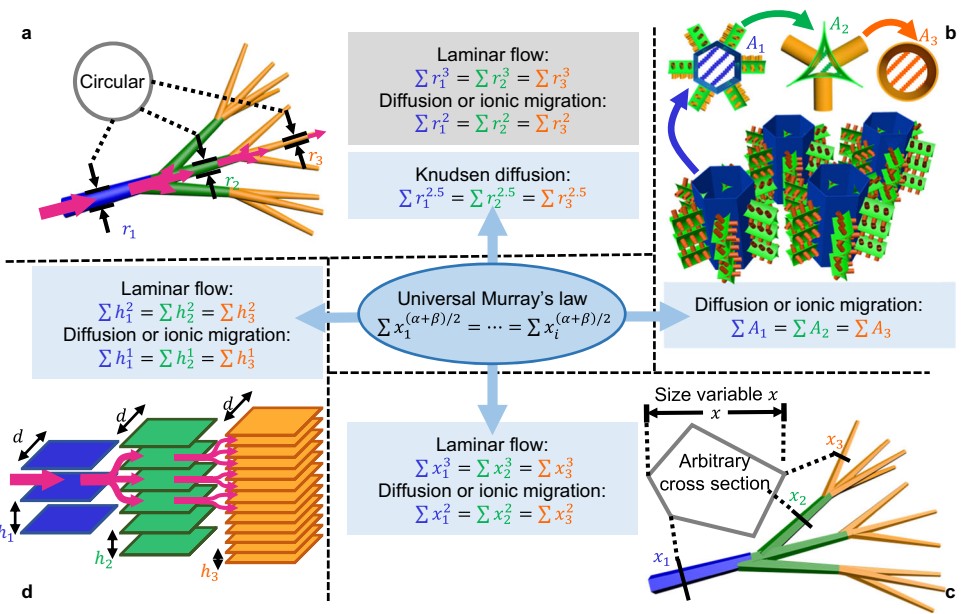

**Fig. 1 | Murray's law in hierarchical structures. a** Schematic illustration of branching columnar tubes and corresponding initial expressions of Murray's law. Here $r_1$, $r_2$, and $r_3$ represent the radii of tubes at different levels. **b** Schematic illustration of demonstrative hierarchical structure in materials with comprehensive pore shapes and corresponding expression of Murray's law. Here $A_1$, $A_2$, and $A_3$ represent the cross-sectional area of pores at different levels. **c** Schematic illustration of hierarchically tubular network with arbitrary shape and corresponding expressions of Murray's law. Here $x_1$, $x_2$, and $x_3$ represent the selected variables of channels at different levels. **d** Schematic illustration of hierarchical lamellar structure and corresponding expressions of Murray's law. Here $h_1$, $h_2$, and $h_3$ represent the heights of planar channels at different levels, $d$ denotes the width of the plates.

We note that Murray's law and its derivation can also be obtained by simply optimising the total resistance in a hierarchical network. For example, the flow resistance of laminar flow $R$ in a hierarchical network could be written as the ratio of pressure difference $\Delta p$ and total volumetric flow rate $Q$[33], $R = \frac{\Delta p}{Q}$. The minimisation of $R$ is equivalent to the maximisation of efficiency, where $\Delta p$ is optimally utilised to drive the flow. Intuitively, the deduction of minimising resistance generates the identical cubic form of Murray's law for laminar flow and the square form for diffusion and ionic migration (for details, see Supplementary Note 2). Compared to the original deduction by investigating power cost, minimising resistance offers a more versatile approach allowing extension of the law to other transfer types like diffusion, where quantifying energy consumption is challenging.

Although previous discussions of Murray's law are all based on tubular networks with circular cross section (Fig. 1a), artificially fabricated channels typically exhibit intricate and diverse shapes that significantly deviate from the perfect circular cross section of a cylinder commonly found in biological networks[10–13,19]. Furthermore, the pore shapes at different levels of a micro or nanostructured material often differ considerably due to the various pore-forming techniques.[10,11,22] Additionally, synthetic materials have large branching numbers owing to the significant size difference in hierarchical pores[5,6]. Figure 1b shows the schematic of a fictitious hierarchically porous network with non-circular and inconstant pore shapes at different levels. The first-level hexagonal pores represent typical self-assembled macropores[11,19] in materials. The second-level pores in the shape of concave circular triangle illustrate the gaps between close-packed nanoparticles, which are often regarded as mesopores in Murray materials[10,13,21]. The geometric difference between the tube models in the current theory and the actual synthetic pores makes it challenging to reliably apply the law to these materials. Additionally, the original Murray's law cannot be expanded to other structures beyond hierarchical tube network or other types of mass transport. To the best of our knowledge, no successful attempts have ever been made to adequately leverage Murray's law in the practical modelling of synthetic hierarchical structures.

## Universal Murray's law

We start the generalisation of a hierarchical network with an arbitrary channel shape to make the derivation universal in terms of pores. For a proper size variable of the channel $x$, if we assume the cross-sectional area has a power law relationship with $x$, we can write $A = k_1 x^\alpha$, where $k_1$ is the linear coefficient, and $\alpha$ is the exponent of $x$ in power function $A$. For example, in a tube with circular cross section ($A = \pi r^2$), when the radius $r$ is selected as variable $x$, $k_1 = \pi$, and $\alpha = 2$. Supposing a potential $\Delta P$ drives a mass transfer process in the network, and assuming the generalised mass flow rate $Q$ also has a power relationship with $x$, written as $Q = k_2 x^\beta \cdot \frac{\Delta P}{l}$, where $k_2$ is the linear coefficient, $l$ is the channel length of a section, and $\beta$ is the exponent of $x$ in power function $Q$. Then, the minimisation of the resistance $R = \frac{\Delta P}{Q}$ gives an equation for the optimal $i$-level hierarchical network (for details, see Supplementary Note 3):

$$\sum x_1^{(\alpha+\beta)/2} = \sum x_2^{(\alpha+\beta)/2} = \cdots = \sum x_i^{(\alpha+\beta)/2} \qquad (4)$$

Equation (4) can readily give optimisation expressions for the well-discussed tubular networks with circular cross section (Supplementary Table 1), conforming to the known results. We name this expression Universal Murray's law, as it represents the most general form of Murray's law to date.

More broadly, we can optimise hierarchical structures with inconstant shapes at different levels. Because of the dissimilar pore shapes, the cross-sectional area $A$ and transfer rate $Q$ of the channels at different levels show separate expressions. As shown in

Supplementary Fig. 1, for an $i$-level hierarchical network, the expressions of the cross-sectional area at each level can be written as $A_1 = k_{1,1} x_1^{\alpha_1}$, $A_2 = k_{1,2} x_2^{\alpha_2}$, $\cdots$, $A_i = k_{1,i} x_i^{\alpha_i}$, with the flow rates $Q_1 = k_{2,1} x_1^{\beta_1} \frac{\Delta P}{l}$, $Q_2 = k_{2,2} x_2^{\beta_2} \frac{\Delta P}{l}$, $\cdots$, $Q_i = k_{2,i} x_i^{\beta_i} \frac{\Delta P}{l}$, where the subscripts 1, 2, $\cdots$, $i$ represent the level number. The Universal Murray's law then transforms to:

$$\sum x_1^{(\alpha_1+\beta_1)/2} : \cdots : \sum x_i^{(\alpha_i+\beta_i)/2} = \sqrt{\frac{\beta_1}{k_{1,1} k_{2,1} \alpha_1}} : \cdots : \sqrt{\frac{\beta_i}{k_{1,i} k_{2,i} \alpha_i}} \qquad (5)$$

The detailed derivation is in Supplementary Note 3.

Additionally, when the flow rate of the mass transfer process is not linear to the potential difference, but can still be written as $Q = k_2 x^\beta \cdot \left(\frac{\Delta P}{l}\right)^\gamma$, where $\gamma$ is the exponent of $\left(\frac{\Delta P}{l}\right)$, our Universal Murray's law gives (Supplementary Note 3):

$$\sum x_1^{\frac{\gamma\alpha+\beta}{1+\gamma}} = \sum x_2^{\frac{\gamma\alpha+\beta}{1+\gamma}} = \cdots = \sum x_i^{\frac{\gamma\alpha+\beta}{1+\gamma}} \qquad (6)$$

Equation (6) can be used to readily optimise the turbulent flow in rough pipes ($\sum r_1^{7/3} = \cdots = \sum r_i^{7/3}$), turbulent flow in smooth pipes ($\sum r_1^{17/7} = \cdots = \sum r_i^{17/7}$), and laminar flow of non-Newtonian liquids following power-law rheology ($\sum r_1^3 = \cdots = \sum r_i^3$), consistent with the reported results (Supplementary Note 4).

Our proposed Universal Murray's law first generalises the original Murray's law for arbitrary hierarchical structures and transfer types, even for a network with dissimilar channel (pore) shapes. These above expressions enable the investigation of the general mass flow principle of optimal networks, independent of the existing models based on specific structures and transfer processes.

## Expanding Murray's law for hierarchical structures in materials

The Universal Murray's law can optimise a more general hierarchical structure and transfer process, accommodating various pore shapes in materials. For example, it extends the optimisation for the well-studied tubular network with circular cross section to the network with non-circular channels as discussed in Supplementary Note 5. Figure 1c schematically illustrates a tubular branching network with an arbitrary close geometric shape while the channel sections at different levels are similar. In this case, the Universal Murray's law proves that the cubic expression of the original Murray's law for laminar flow, $\sum x_1^3 = \sum x_2^3 = \cdots = \sum x_i^3$, and the square form for diffusion, ionic migration, or electron transportation, $\sum x_1^2 = \sum x_2^2 = \cdots = \sum x_i^2$ are still valid for tubular networks with any channel shape.

More generally, for hierarchically porous materials with different shapes at different levels (Fig. 1b), we note that for diffusion, ionic migration, or electron transportation, a proportionate relationship exists between their transfer rate and cross-sectional channel area, $Q \propto A$, independent of the channel shape. This is consistent with Pouillet's law and Ohm's law for ionic migration and electron transportation $Q = \sigma A \cdot \frac{\Delta V}{l}$ (where $\sigma$ is the conductivity and $\Delta V$ is the potential difference) and Fick's law for diffusion $Q = DA \cdot \frac{\Delta C}{l}$, (where $D$ is the diffusion coefficient and $\Delta C$ is the concentration difference). If $A$ is directly chosen as the size variable, $\alpha = 1$ and $\beta = 1$, the Universal Murray's law gives:

$$\sum A_1 = \sum A_2 = \cdots = \sum A_i \qquad (7)$$

Thus, for diffusion or ionic migration which are most commonly found in the applications of porous materials, our Universal Murray's law shows that the above optimisation equation holds irrespective of the exact pore shape in hierarchical materials. This expression proves that structural optimisation can be directly linked to the pore's cross-sectional area or the normalised pore size obtained by gas sorption

analysis[10–12,18,19]. Beyond the above transfer types, hierarchically porous materials with different pore shapes can be optimised by Equ. (5).

## Optimising unexplored structure and transfer by Universal Murray's law

Beyond unifying and expanding the known expressions of the original Murray's law, our Universal Murray's law can also optimise unexplored transfer types and hierarchical structures. For instance, Knudsen diffusion describes gas diffusion in mesopores with pore sizes comparable to mean free path, with Knudsen diffusion coefficient scaling as $D_k \propto r$[34]. Thus, in a tubular network (Fig. 1a), the molecular flow rate across the channel according to Knudsen formula is linear to the cube of the radius: $Q \propto r^3$[35]. We can write the optimal hierarchically tubular structure for this case using Universal Murray's law:

$$\sum r_1^{2.5} = \sum r_2^{2.5} = \cdots = \sum r_i^{2.5} \qquad (8)$$

We can also optimise planar material structures with the Universal Murray's law. As shown in Fig. 1d, this type of hierarchical structure fractally divides into only one dimension, while the tubular structure branches into two dimensions. Let us consider the plates at different levels in Fig. 1d to have a fixed width $d$, which is significantly larger than the channel height $h$, and thus, can be regarded as a 2D infinite planar structure. Consequently, if we select the height of the channels between parallel plates $h$ as the size variable for optimisation, the relationship between $A$ and $h$ is $A \propto h^1 (\alpha = 1)$. Meanwhile, the volumetric flow rate of a laminar flow between the parallel plates can be written as $Q = \frac{d}{12\eta} \cdot h^3 \cdot \frac{\Delta p}{l}$ ($\beta = 3$)[36], where $\eta$ represents the fluid viscosity, $\Delta p$ is the pressure difference, and $l$ is the channel length. According to our Universal Murray's law, the optimised planar structure should satisfy $\sum h_1^2 = \sum h_2^2 = \cdots = \sum h_i^2$ for laminar flow. The corresponding expression of $Q$ gives $\sum h_1 = \sum h_2 = \cdots = \sum h_i$ for diffusion and ionic migration, as shown in Supplementary Table 2.

The aforementioned discussions in Knudsen diffusion and planar structure demonstrate that our Universal Murray's law can concisely express the optimal design of unexplored hierarchical networks and transfer types. This general theory shows great potential in optimising synthetic porous structures for various application scenarios. In the following sections, we offer two experimental examinations of the Universal Murray's law.

## Unidirectional and bidirectional freeze-cast GOA

Graphene oxide aerogel is chosen to construct the hierarchical structures to verify our proposed Universal Murray's law, because of its advantageous characteristics of high porosity, appropriate pore size, and the feasibility of adjusting the pore size and shape. Freeze-casting method, also known as the ice templating[37], allows for the preparation of various shapes of GOA with adjustable pore size within a certain range. The GOA-based Murray structures in the following sections are constructed by this preparation method. The density of the as-prepared GOA (see Methods) is low to 25 mg cm$^{-3}$ (Supplementary Fig. 2). This corresponds to a calculated porosity of 98.2%[38]. The pore wall thickness in the aerogel is negligible, and is not considered in the calculations. The reported pore diameters of freeze-cast aerogels range from 10 to 240 microns, far above the upper limit of the nanofluidic channel (<100 nm)[39]. Hence, the electrical double layer at the GO surface does not significantly influence the flow[40].

As shown in Fig. 2a, we use the unidirectional freeze-casting method to prepare the vertically porous GOA. This method applies a temperature gradient in the vertical direction to the samples on a horizontal freezing platform of copper. Temperature changes at different heights during the unidirectional freeze-casting demonstrate the vertical temperature gradient and the bottom-to-top ice growth (Supplementary Fig. 3a). While freezing, random nucleation of ice crystals first appears and grows on the cooling surface, forming

multiple, small-sized domains at the interface of the GO dispersion and the copper platform[41]. The ice crystals then elongate in parallel along the freezing direction under the restriction of neighbouring crystals. The columnar ice crystals, serving as heterogeneous templates, extrude the assembled materials and create a framework during their solidification. Subsequent sublimation removes this ice template, retaining a refined porous structure. The vertical pores in GOA frozen at different temperatures are shown in Fig. 2b–d and Supplementary Fig. 4. The top-view and side-view (Supplementary Fig. 5) of freeze-cast aerogel illustrate dense and aligned columnar pores, originating from vertically-grown ice crystals through unidirectional freeze-casting. As a comparison, the GOA frozen by liquid nitrogen demonstrates an almost isotropically porous microstructure (Supplementary Fig. 6), where the twisty and short pores do not show a specific preference for prolonged direction due to omnidirectional freezing. According to the basic principle of crystallography[42], the ice crystallization rate increases with decreasing freeze-casting temperature, resulting in smaller ice grain sizes and vertical pores in the resulting material. This phenomenon is significant in the fabrication of porous materials by freeze-casting, as it allows for adjustment of the resulting pore structure. Consequently, the vertical pores significantly shrink as the freezing temperature is reduced from −20 °C to −70 °C, revealed in the top-view scanning electron microscope (SEM) images (Fig. 2b–d).

We also quantitatively analyse the average pore size of the frozen samples using a bespoke image processing program (see Methods and Supplementary Software). As shown in Supplementary Fig. 7a, the program recognises and masks the pores at the sectional image by adjusting contrast. The program then calculates and averages the pore sizes. Compared to manual recognition[39], this approach is more objective and reliable.

As shown in Fig. 2e, the pore size is closely related to the freezing temperature with a uniform diameter distribution (Supplementary Fig. 8). The average pore diameter of liquid nitrogen-frozen GOA at −196 °C is 6.23 µm. The pore sizes of directionally freeze-cast GOA range from 7.85 µm to 39.8 µm, corresponding to the freezing temperature increase from −80 °C to −10 °C. However, the temperature-size relation is not linear. The average pore diameter rises slowly at low temperatures but increases rapidly near 0 °C. This is because along with the increase in freezing temperature, the ice nucleation rate decreases non-linearly, while the growth rate increases initially and then reduces[39].

Figure 2f illustrates the bidirectional freeze-casting method for constructing the planar structure. When a polydimethylsiloxane (PDMS) wedge with a certain angle is inserted between the cooling stage and the sample being freeze-cast, an additional horizontal temperature difference (Fig. 2f) is applied in the deposited GO due to the low thermal conductivity of PDMS[41,43]. As shown in Supplementary Fig. 3b, the sequence of cooling and freezing at different positions reveals the temperature gradients in two directions during the bidirectional freezing. The bidirectional temperature difference causes the ice crystals to grow both horizontally and vertically, resulting in a layered structure after freeze-drying. The aerogel, which is bidirectionally freeze-cast at a relatively high temperature, shows a regular lamellar architecture in the top-view (Fig. 2g and Supplementary Fig. 9) and side-view section (Supplementary Fig. 10). The parallel GO walls align with the plane of two temperature gradients. The lamellar GOA samples frozen at lower temperature illustrate a narrower layer spacing and a more disordered alignment (Fig. 2h–i) due to the rapid ice growth during freezing. Similarly, we developed another image processing program to calculate average layer spacing in lamellar GOA frozen at different temperatures (see Methods for details and Supplementary Fig. 7b). Figure 2h demonstrates that the average layer spacing of lamellar GOA increases from 30.5 µm to 97.1 µm when the bidirectional freeze-casting temperature rises from −50 °C to −10 °C.

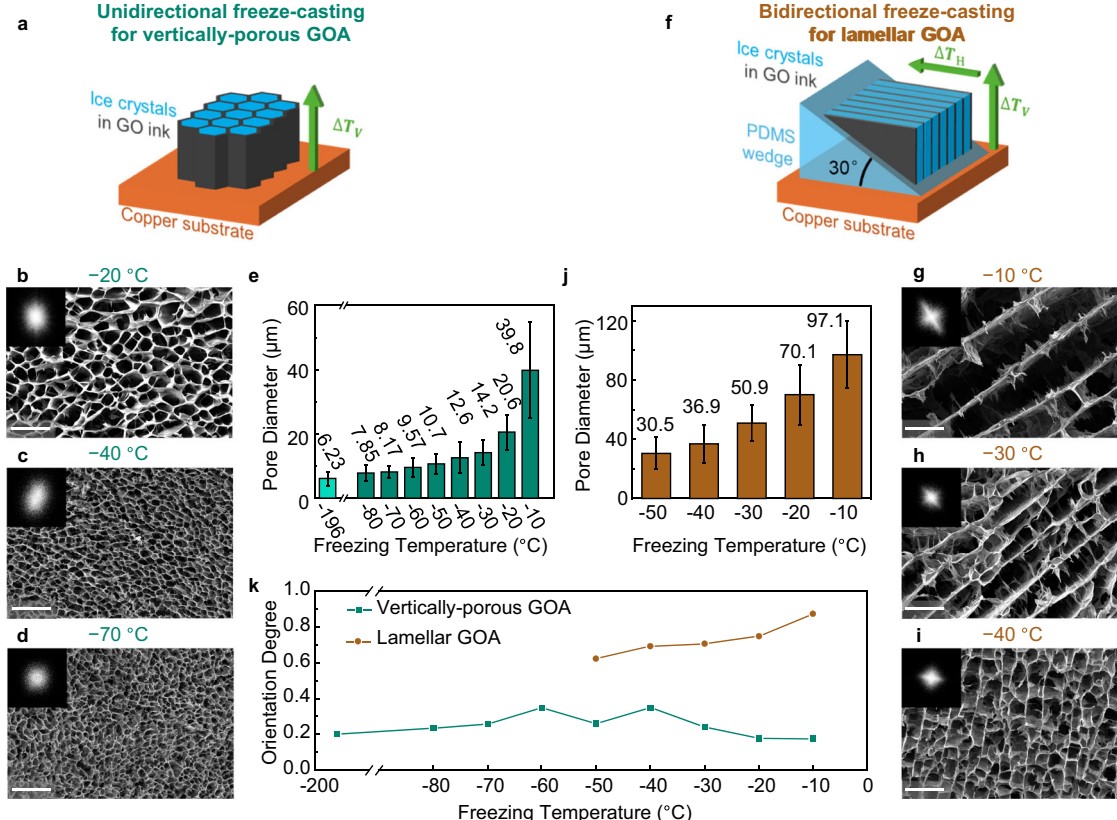

**Fig. 2 | The structure of unidirectional and bidirectional freeze-cast graphene oxide aerogel (GOA). a** Schematic illustration of unidirectional freeze-casting method. Here $\Delta T_V$ denotes the temperature difference in the vertical direction. (**b-d**) Top-view SEM images of unidirectional freeze-cast GOA frozen at (**b**) −20 °C, (**c**) −40 °C, and (**d**) −70 °C. Insets: Fourier transform images. **e** The average pore size of unidirectionally freeze-cast GOA frozen at different temperatures and GOA frozen by liquid nitrogen. **f** Schematic illustration of bidirectional freeze-casting method. Here $\Delta T_V$ denotes the temperature difference in the vertical direction and $\Delta T_H$ represents the temperature difference in the horizontal direction. **g-i** Top-view SEM images of bidirectionally freeze-cast GOA frozen at (**g**) −10 °C, (**h**) −30 °C, and (**i**) −50 °C. Insets: Fourier transform images. **j** Average layer height of bidirectional freeze-cast GOA at different temperatures. **k** Orientation degree of unidirectionally and bidirectionally freeze-cast GOA. Scale bars: 100 $\mu$m. All error bars represent standard deviations of pore size calculated by the corresponding image processing programs (see Methods).

Fourier transforms of the raw SEM images (Fig. 2b–d, g–i, insets) show the orientation of the GO structure in the cross section[41]. The unidirectional freeze-cast GOA (Fig. 2b–d, insets) and liquid-nitrogen frozen GOA (Supplementary Fig. 6, inset) have close to circular Fourier transforms, implying there is no specific orientation along the cross-section. In contrast, a clear alignment is observed in the top-view section of lamellar GOA prepared by bidirectional freeze-casting (Fig. 2g–i, insets). A pore shape fitting program (Supplementary Software), similar to what we used for the aforementioned pore measurement, is also used to estimate a quantitative value to describe the pore orientation. From 0 to 1, orientation degree denotes the fully random distribution of pore orientation to perfect alignment. In Fig. 2k, bidirectionally freeze-cast lamellar GOA possesses a higher orientation degree from 0.88 to 0.62, declining with the decrease of freezing temperature. In contrast, the disordered, vertically porous GOA or the one frozen by liquid nitrogen show a lower orientation degree, ranging from 0.18 to 0.30.

**Examination of Universal Murray's law in hierarchical structures**
Thus far, comparative experiments on the mass transfer superiority of Murray materials have never been successfully demonstrated. Current studies compare materials with different levels of hierarchy and show that the materials with higher levels of hierarchy exhibit superior performance, for example, those with three levels of macro-, meso-, and micropores outperform those with only two levels of meso- and micropores[10,12,19]. These comparisons only demonstrate the benefit of

introducing an additional level of hierarchy, rather than mass transfer improvement of structural optimisation based on Murray's law. Therefore, the verification of optimal mass transfer performance in Murray materials requires comparison between samples following or deviating from the law but with the same level of hierarchy.

According to the equations of Murray's law, to construct an optimal *i*-level hierarchical structure, it is necessary to precisely control either the size or the number of channels in at least $(i-1)$ levels. For freeze-cast GOA, the gap size can be adjusted through the tuning of the freezing temperature. However, it is impractical to accurately fine-tune the channel size by this process. We therefore tune the channel numbers by shaping the bulky aerogel when constructing GOA-based Murray structures.

Laminar flow is the most obvious transport process in living organisms[23,30], and thus, it is the first and most widely-discussed transfer type for Murray's law. Towards this end, we first survey the optimal planar structure for laminar flow obeying the Universal Murray's law in GOA. In a 3-level rectangular mould, we derive the expression for laminar flow in the planar structure based on Universal Murray's law, $n_1 h_1^2 = n_2 h_2^2 = n_3 h_3^2$ (Supplementary Table 2). This equation can be rewritten as: $H_1 h_1 = H_2 h_2 = H_3 h_3$, where $H_1$, $H_2$, and $H_3$ represent the section height at different levels (Supplementary Note 6). Lamellar GOA frozen at −10 °C, −30 °C, and −50 °C are applied to construct the hierarchical structure with the maximised size difference between the levels. Therefore, the section heights need to satisfy $H_1 : H_2 : H_3 = 0.524 : 1 : 1.67$, according to their average layer

spacing of 97.1, 50.9, 30.5 $\mu$m (Fig. 2j). Note that, to avoid excessive concentration of the flow current at the pipe centre, we ensure smooth transitions between the sections as demonstrated and discussed in Supplementary Fig. 11.

We also prepare other hierarchically planar pipes that deviate from Murray's law and obey the conservation $\sum h_1^x = \sum h_2^x = \sum h_3^x$ for exponent values of $x = 1, 1.5, 2.5, 3$, with the same channel length, width, and total volume. These channels are compared with the aforementioned Murray structures following $\sum h_1^2 = \sum h_2^2 = \sum h_3^2$. We then measure the pressure drop along these hierarchical pipes for both water and air flow at different flow rates (Supplementary Fig. 12). All the tests discussed in this section have laminar flow under our experimental conditions (see Methods). We show the plotted flow resistance calculated from the pressure drop in Fig. 3a for water flow and Fig. 3b for air flow with the exponent value $x$ as the horizontal axis. The aerogel following Murray's law demonstrates the smallest resistance for laminar flow. Furthermore, as the hierarchical network deviates from Murray's law, $\sum h_1^2 = \sum h_2^2 = \sum h_3^2$, resistance increases.

We then use scaled-down models for flow simulation (Supplementary Fig. 13). Note that the simulation of full-scale models is unnecessary and impractical because of the considerable gap between the size of bulky aerogel samples and the pores. The simulation results show the same U-shape curve, with the lowest flow resistance point matching the synthetic Murray materials (Fig. 3a, b). Additionally, the deduction of this theory relies on the even distribution of the flow within individual channels of each section. The simulation (Supplementary Fig. 13) illustrates that the flow distribution is roughly uniform in the aerogel, satisfying this assumption.

The examination of the planar structure for optimal laminar flow is convincing evidence for the establishment of Universal Murray's law. To the best of our knowledge we, for the first time, expand Murray's law into a hierarchically planar structure and experimentally confirm it. Both the experiments and simulation demonstrate that the isochoric lamellar GOA structures achieve minimised resistance for laminar flow when following the corresponding expression of Universal Murray's law. We also show that

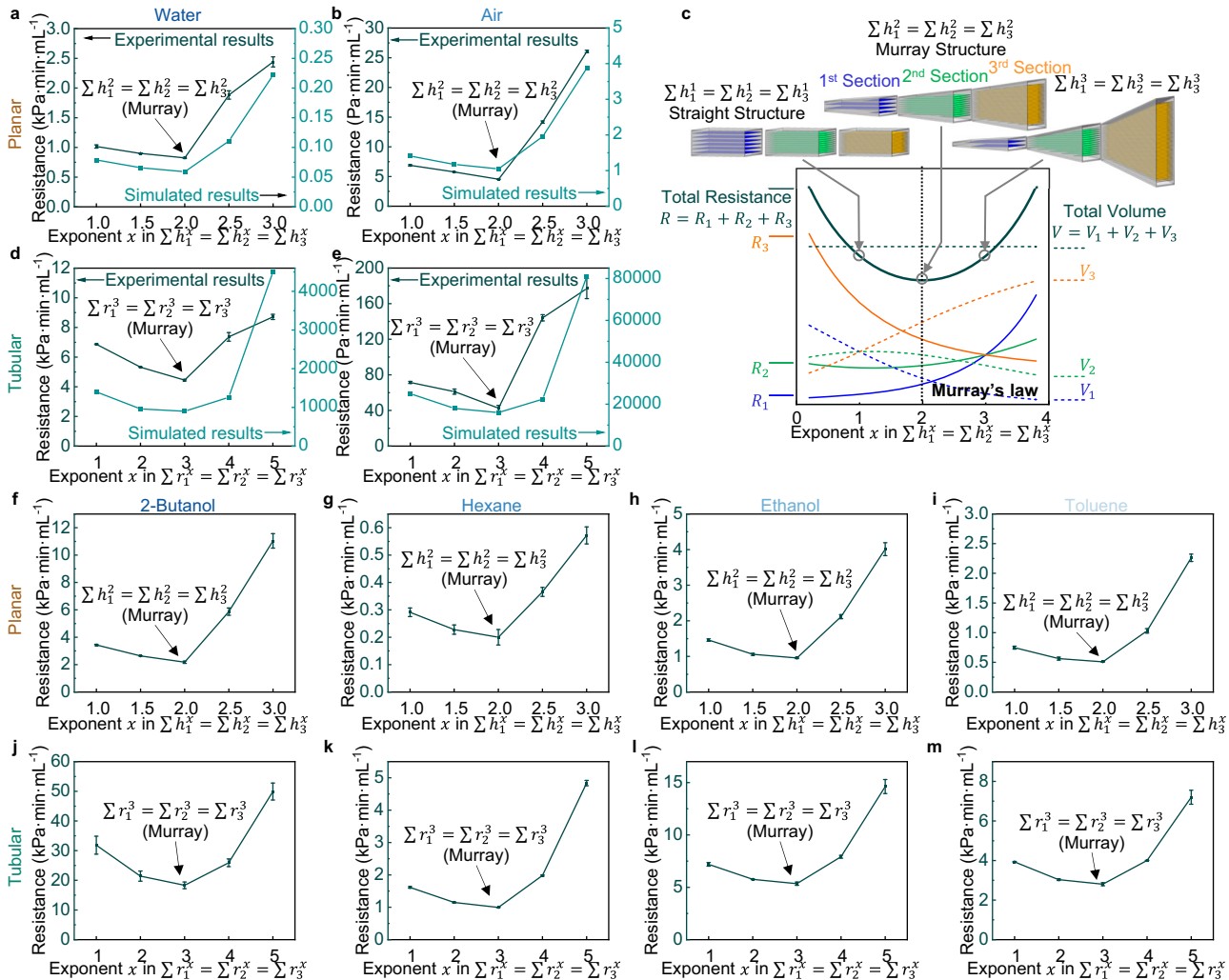

**Fig. 3 | The experimental and simulation validation of Universal Murray's law.**
**a**, **b** The experimental flow resistance of (**a**) water and (**b**) air in hierarchical lamellar GOA with a fixed total volume and corresponding simulated resistance in scaled-down models. **c** Changes in the estimated laminar flow resistance and section volume with exponent $x$ in the conservation. Here $R_1$, $R_2$, $R_3$, and $R$ represent the estimated resistances of different sections and the total resistance. $V_1$, $V_2$, $V_3$, and $V$ represent the volumes of different sections and the total volume. **d**, **e** The

experimental flow resistance of (**d**) water and (**e**) air in hierarchically tubular GOA and corresponding simulated results in scaled-down models. **f**–**i** The experimental flow resistance of (**f**) 2-butanol, (**g**) hexane, (**h**) ethanol, and (**i**) toluene in hierarchical lamellar GOA. **j**–**m** The experimental flow resistance of (**j**) 2-butanol, (**k**) hexane, (**l**) ethanol, and (**m**) toluene in hierarchical tubular GOA. The error bars arise from the linear fit of pressure drop to the flow rate as discussed in Methods.

the deviation from this principle leads to reduction of mass transfer performance, as enshrined in Murray's law.

We note that the layered structure with a larger layer spacing should have a smaller flow resistance of the same shape, as the flow resistance of a section can be rewritten as $R \propto \frac{1}{nh^3} = \frac{1}{Hh^2}$, where $n$ represents the number of channels, $h$ is the channel height, and $H$ is the section height. The flow resistance of lamellar GOA frozen at different temperatures (Supplementary Fig. 14) and the simulation results (Supplementary Fig. 15) also confirm this trend. Therefore, in the straight structure following $\sum h_1^1 = \sum h_2^1 = \sum h_3^1$, the flow resistance of the latter sections should be larger than the former $R_1 < R_2 < R_3$, because $h_1 > h_2 > h_3$ and $H_1 = H_2 = H_3 = \sum h$. For other structures, with the rising of exponent $x$ in the conservation formula, the volume of the whole structure gradually transfers from the front to the latter sections; Fig. 3c. This shape change would reduce the resistance of the third section and raise the resistance in the first two sections; Fig. 3c. Additionally, it also increases the total surface area of hierarchical structure, because the high-level section with smaller pores has a higher specific surface area. The U-type resistance curve in Fig. 3a–c implies that in this process, resistance reduction in the third section is initially dominant, followed by a resistance increase in the first two sections. The two aforementioned influences are equal when obeying Murray's law, showing the lowest resistance. Consequently, with a constrained total volume, Murray's law can also be described as a principle that appropriately distributes a larger volume into the high-level sections of smaller channel sizes and higher resistances, such that it balances the resistances of different sections to minimise the total resistance.

The structural optimisation based on Murray's law in hierarchically tubular pipes prepared by vertically porous GOA also supports this observation. Although the channels in unidirectionally freeze-cast GOA are more like close-packed polygonal tubes rather than the cylinders assumed in the original Murray's law (Fig. 2b–d), the Universal Murray's law allows optimisation for this type of non-circular pores. Similarly, using vertically-porous GOA frozen at $-20\,^\circ$C, $-40\,^\circ$C, and $-70\,^\circ$C, we construct and compare hierarchical channels following $\sum r_1^x = \sum r_2^x = \sum r_3^x$, where exponent $x = 1, 2, 3, 4, 5$ (Supplementary Note 6). We also improve the pipeline to a smooth conical shape for better flow distribution in the channel (Supplementary Fig. 16). As shown in Fig. 3d, e and Supplementary Fig. 17, with the same volume, the pipe obeying Murray's law ($\sum r_1^3 = \sum r_2^3 = \sum r_3^3$) achieve minimal resistance for laminar fluid flow both in experiments and in simulation of the scaled-down models (Supplementary Fig. 18). The resistance increases notably when the exponent $x$ moves away from 3, referring to pipes gradually deviating from the optimal Murray structure. As a more classic and frequently discussed model[23,27,28], these results of the tubular structure further validate the Universal Murray's law in materials by both experimentation and simulation. Besides laminar flow, the planar and tubular structures following the corresponding Universal Murray's law for diffusion also exhibit optimum diffusion efficiency, as shown in the simulation results in Supplementary Fig. 19.

Laminar fluid flow is important in industrial production, such as the catalytic reaction of organic solvents. Since our deduction of the Universal Murray's law does not consider the type of fluid, the optimisation is also expected to be universally applicable to other fluids under laminar flow. To verify this, we measure the laminar flow resistance in several common and representative organic solvents, including high viscosity solvents such as 2-butanol (Fig. 3f, j), low viscosity solvent hexane (Fig. 3g, k), polar solvents ethanol (Fig. 3h, l) and 2-butanol, and non-polar solvents toluene (Fig. 3i, m) and hexane. The experiments on these solvents show that both the hierarchically planar and tubular GOA reach minimal resistance when obeying Universal Murray's law. The experimentally obtained U-type curves again effectively demonstrate the universality of this principle.

## Optimising hierarchical GOA-based gas sensor by Universal Murray's law

To demonstrate the practical applicability of Murray's law, we conceive a GOA-based gas sensor to measure nitrogen dioxide ($NO_2$) flowing through it, and then optimise the hierarchical structure for gas flow using the Universal Murray's law. The gas sensor is prepared by $SnO_2$ quantum dots (QDs) decorated GOA as shown in Fig. 4a[44]. $SnO_2$ QD-decorated GO ink is synthesised through a surfactant-assisted hydrothermal growth process (see Methods for details). Then, the decorated ink is unidirectionally freeze-cast at $-20\,^\circ$C, $-40\,^\circ$C, and $-70\,^\circ$C to construct the hierarchically porous aerogels. Transmission electron microscopy (TEM) images of as-prepared GO ink (Fig. 4b–d) demonstrate uniformly distributed $SnO_2$ QDs on GO sheets, with size smaller than twice the exciton Bohr radius (2.7 nm) of $SnO_2$[44]. The lattice fringes of 2.5 and 3.3 Å in Fig. 4c correspond to 101 and 110 planes of $SnO_2$, respectively. X-ray diffraction (XRD) patterns in Supplementary Fig. 20 of $SnO_2$ QD-decorated GO and $SnO_2$ QDs can be attributed to the tetragonal phase of $SnO_2$ (JCPDS card no. 41-1445). $SnO_2$ QD-decorated GO shows an additional characteristic diffraction peak of GO (001)[45]. As shown in Fig. 4e–g, after unidirectionally freeze-casting, $SnO_2$ QD-decorated GOA forms consistent vertical pores (20.6 $\mu$m at $-20\,^\circ$C, 12.6 $\mu$m at $-40\,^\circ$C, and 8.17 $\mu$m at $-70\,^\circ$C) with pure GOA (Fig. 2e) within the measurement error range. Note that at the high GO concentrations we used (25 mg mL$^{-1}$), the addition of the quantum dots does not significantly affect the pore size of freeze-cast aerogel[46].

Without considering any structural design principles, we first intuitively conceive a hierarchical gas sensor as a straight cylinder with three levels of sections (Fig. 4a). The hierarchy of the materials in these three sections offers both the benefits of efficient airflow and large active surface area. This sensor structure follows the conservation $\sum r_1^2 = \sum r_2^2 = \sum r_3^2$. With the same total volume of $30\pi$ mm$^3$ and length of 10 mm, we now adjust the average diameters of each section to $D_1 : D_2 : D_3 = 0.799 : 1 : 1.23$ to construct the Murray structure obeying $\sum r_1^3 = \sum r_2^3 = \sum r_3^3$. This simple shape adjustment based on Murray's law shortens the response times $\tau_{res}$ and recovery times $\tau_{rec}$ by from 8.6% to up to 18.2% in the gas sensing of 1 ppm $NO_2$, $NH_3$, and $CH_2O$ (Fig. 4h and Supplementary Fig. 21). This improvement degree in response and recovery of gas sensing can be explained by the improved mass transport associated with the 12.3% reduction in the air flow resistance, as indicated in the simulation of the scaled-down models (Fig. 4i). By tailoring the shape of the GOA following Murray's law, we reinforce the dynamic response and recovery of the sensor due to the fluid transport improvement. Therefore, structural optimisations based on Murray's law can considerably strengthen the performance of porous materials by simply adjusting the macroscopic shape or pores, without changing the material's chemical composition. For hierarchically porous materials used for applications relying on mass transfer, such as catalysis[47], sensing[48,49], energy storage[4], and environmental protection[50], Murray's law can therefore offer significant performance improvement.

## Discussion

We have proposed an extension of Murray's law for synthetic nanostructures regardless of channel shape and experimentally demonstrated its validity. Starting from minimising the resistance of a general transport process, we extend the original expression to a common mathematical form as the Universal Murray's law. For diffusion, ionic migration and electron transportation in arbitrary branching networks, the optimised law can be transformed into an equation of the sum of corresponding cross-sectional area. This result provides a rigorous theoretical framework for the most hierarchically porous materials with non-circular shapes. Additionally, we discuss two scenarios, Knudsen diffusion and planar shape, to demonstrate the value of Universal Murray's law towards unexplored transfer type and structure. We construct planar and tubular structures by freeze-cast

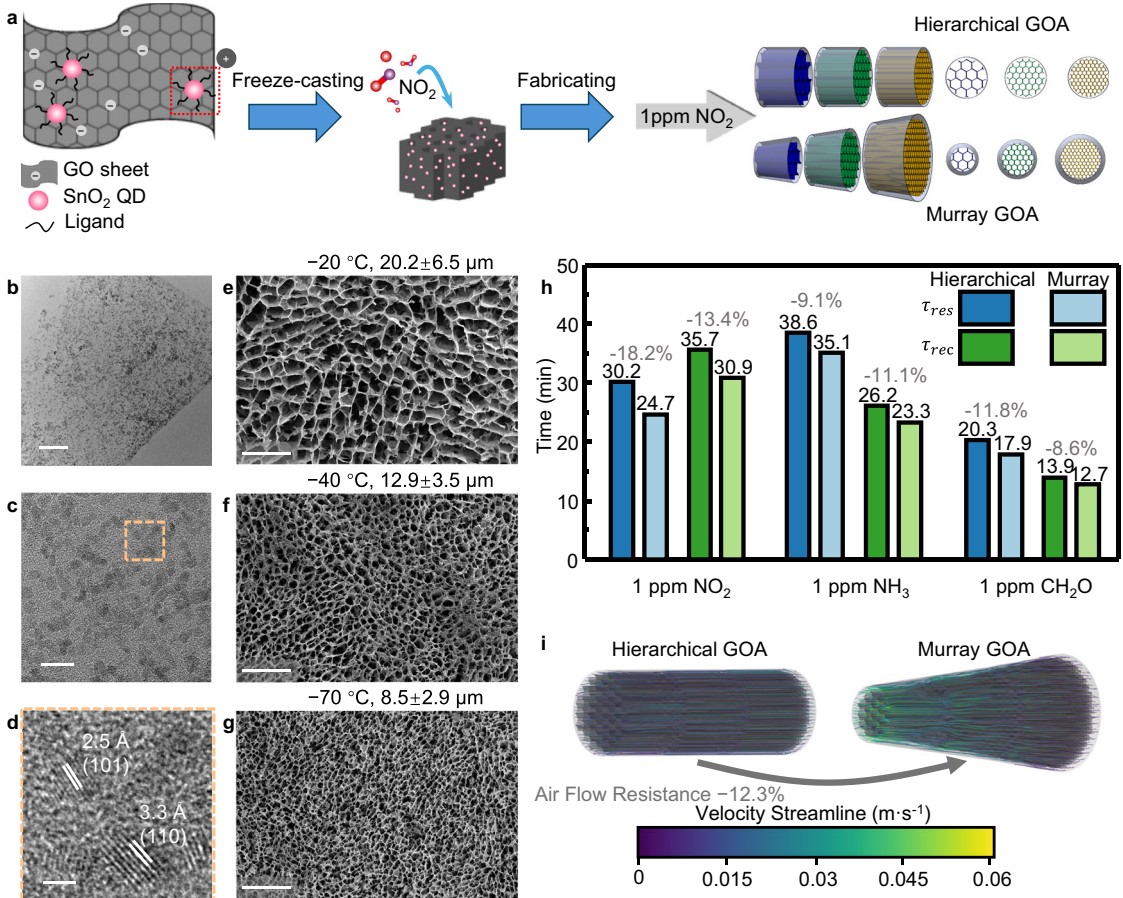

**Fig. 4 | Optimising tubular GOA-based gas sensor by Murray's law. a** Schematic illustration of the synthesis of SnO₂ quantum dot (QD)-decorated graphene oxide aerogel (GOA) and the assembly of hierarchical gas sensor. **b**–**d** TEM images of SnO₂ QD-decorated GO ink. Scale bars: (**b**) 100 nm, (**c**) 10 nm and (**d**) 2 nm. **e**–**g** Top-view SEM images of unidirectional freeze-cast GOA frozen at (**e**) −20 °C, (**f**) −40 °C, and (**g**) −70 °C. Scale bars: 100 $\mu$m. **h** Response time ($\tau_{res}$) and recovery time ($\tau_{rec}$) of SnO₂ QD-doped GOA in the hierarchical straight pipe and the pipe optimised by Murray's law to 1 ppm nitrogen dioxide, ammonia, and formaldehyde. **i** Air flow simulation in the scaled-down models of hierarchical and Murray GOA. The standard deviations of pore size are calculated by the corresponding image processing programs (see Methods).

GOA to verify the validity of the Universal Murray's law in materials. The hierarchical aerogels obeying the optimisation equation show minimal resistance for laminar fluid flow, while others deviating from the principle exhibit increased resistance. We also demonstrate that a simple adjustment of the macroscopic shape guided by this law could significantly improve the mass transfer performance in sensors. Our work establishes a sound theoretical foundation for synthetic Murray materials and may inspire structural design of porous systems in a variety of applications benefiting from optimal mass transport.

## Methods

### Preparation of graphene oxide aerogel

Graphene oxide aerogels are fabricated according to our previously reported method[51]. First, GO dispersion of 25 mg mL⁻¹ concentration is prepared by mixing non-exfoliated GO paste (Sigma-Aldrich) in DI water and stirring for 4 h. The dispersion is then mixed with 160 mM ascorbic acid (Acros Organics) and heated at 60 °C for 1 h to gelate the dispersion through the partial reduction of GO[51]. The as-prepared viscous ink is next extruded into 3D-printed PLA moulds to control the shape of the final samples. After being freeze-cast or frozen in liquid nitrogen for 10 min, the samples are freeze-dried overnight in a freeze drier (LyoQuest, Telstar). The excess ascorbic acid and other soluble impurities are washed away by water and an additional freeze-drying step to obtain GOA.

The freeze-casting method utilises a custom-built freezing device with a copper freezing platform as cold source. In unidirectional freeze-casting, the samples at room temperature are directly placed on the precooled platform. The platform is precisely maintained at a specific temperature for 30 min. As for bidirectional freeze-casting, the samples are placed on a precooled 30° PDMS wedge on the platform for 30 min.

### Preparation of SnO₂ QD-decorated aerogel

SnO₂ QD-decorated GOA for room-temperature gas sensing is synthesised following our previous publication[44]. In a typical process, SnO₂ precursor is first synthesised by dissolving 4 mM of tin chloride pentahydrate (SnCl₄·5H₂O, Sigma-Aldrich) and 4 mM of 6-aminohexanoic acid (AHA, Sigma-Aldrich) in 30 mL DI water, followed by 5 min of sonication. The precursor and 10 mL GO dispersion is then hydrothermally heated at 140 °C for 3 h. After cooling down, the resultant sample is centrifuged at 4000 rpm for 10 min and washed with DI water for 3 times. The SnO₂ QDs are prepared in the same way without the GO dispersion. The precipitate is redispersed into DI water to a concentration of 15 mg mL⁻¹. It is next mixed with 28 mM ascorbic acid and 50 mM copper chloride (CuCl₂, Sigma-Aldrich). The resultant sample is then heated at 60 °C for 30 min to prepare the ink for extrusion. XRD patterns were measured from 5° to 90° by a Bruker D8 Advance powder X-ray diffractometer with CuKα (1.54 Å) radiation. High-resolution TEM (HRTEM, Tecnai F20) is performed to

characterise the QD-decorated sample. The extruded architecture in the mould is then freeze-cast, freeze-dried overnight, and heated at 60 °C for 5 h. The decorated GOA is soaked in 100 mM CuCl$_2$ solution for 1 h and washed with DI water for 1 h 3 times to introduce additional surface doping. A second freeze-drying process removes water from the aerogel, resulting in SnO$_2$ QD-decorated GOA for gas sensing.

## Pore recognition and measurements

After freeze-casting, the frozen samples are horizontally broken at 5 mm height before freeze-drying. SEM (FEI Magellan 400 SEM) is performed to observe the sample cross-section. Fourier transform images are obtained by ImageJ on the inverted figures. The image processing programmes uploaded into Supplementary Software measure the average pore size, layer spacing, orientation degree, and pore aspect ratio. For vertically porous GOA, the code first enhances the contrast of top-view SEM images by gamma conversion and Otsu's thresholding. Then the bright regions representing GO walls are flood-filled and blurred by the box filter and median filter. After masking the holes covered by bright region contour and processing with morphological opening, the average radius of recognised pores is calculated based on their area. For lamellar GOA prepared by bidirectional freeze-casting, the layered holes are first identified with a similar process, and the smallest 10% are removed. They are then fitted with ellipses where the average length of short axes is considered as layer spacing. The porous structures are fitted with ellipses and the rotation angles are calculated to measure the orientation degree. The orientation degree is defined as the average relative deviation of the angle distribution from the uniform distribution. This parameter denotes the alignment degree of the ellipsoids. The orientation degree is close to 1 for perfectly aligned holes and 0 for large numbers of randomly orientated ellipses.

## Flow resistance measurement

The hierarchically planar structures are designed based on the corresponding equations, under the constraints of a total volume of 1300 mm$^3$, a pipe width of 10 mm, and a pipe length of 30 mm. Similarly, the hierarchically tubular structures are designed under the restrictions of a total volume of 90$\pi$ mm$^3$ and a fixed length of 30 mm. The as-designed hierarchical structure of freeze-cast GOA is then connected to a syringe. The volumetric flow rate of fluids through the pipe is controlled by a syringe pump (ALADDIN-220, WPI Ltd.). A digital pressure gauge (Digitron 2021P) is parallelly connected to the tubes' inlet and outlet which spontaneously measures the pressure difference between the two sides. The resistance of laminar flow in the hierarchical GOA is calculated by linearly fitting the pressure difference as the function of the volumetric flow rate. For single-phase pipe flows, the flow tends to be laminar when its Reynolds number is less than 2000, $R_e = \frac{uL}{v} < 2000$, where $u$ is the flow rate, $v$ is the kinematic viscosity of the fluid, and $L$ is the characteristic linear dimension[52]. In the scenarios discussed in this paper, the calculated Reynolds number is much smaller than 2000 and generally smaller than 50, as the diameter of the pores is in the micrometre regime and the flow rates are relatively low. Therefore, all the fluid flow in the experiments can be regarded as laminar flow.

## Simulation

The numerical simulations are performed using the commercial CFD package of ANSYS Fluent software to solve the fluid flow Navier-Stokes equations inside the models. For the simulation of planar structure, the models are 14 times smaller in length and height than the bulky aerogels used in the experiments. The simulation models for tubular structure are 60 times smaller in length, width and height. The models for the aerogel gas sensor are 30 times smaller in length, width and height. The freeze-cast pore size or channel height in these models is consistent with the measured results. The computational mesh roughly has 70,000 cells for planar structure, and 1,700,000 cells for tubular structure, and the solutions are seen as converged when the residuals reach 10$^{-8}$. Second-order discretisation methods are used throughout the simulations. All the simulations assume laminar flow and smooth wall conditions. The solutions show good convergence characteristics independent of grid sizes and residual values.

## Gas sensing measurements

Gas sensing measurements are conducted in the Kenosistec gas characterisation system. Two mass flow controllers are used to control the flow of dry air and the target gases. A total gas flow of 500 sccm is supplied towards the inlet of GOA sensor to form an even gas flow in the aerogel. Before the measurement, the sensors are stabilised in dry air for 2 h. The sensor resistance is measured at a fixed voltage. All experiments are carried out at 25 °C and atmospheric pressure. The response is defined as the fractional change of sensor's resistance, $\frac{|R_g - R_a|}{R_g} \times 100\%$, where $R_g$ denotes the resistance with target gas, and $R_a$ is the stabilised resistance in ambient air. The response time is defined as the time to reach 90% of the maximum response after the exposure to target gas. The recovery time is defined as the time spent for the response to return to 10% above the baseline after the removal of the target gas.

## Data availability

The data generated in this study are provided in the Source Data file. They have also been deposited in the Figshare database under accession code https://doi.org/10.6084/m9.figshare.25159421. Source data are provided with this paper.

## Code availability

All relevant codes are available at the supplementary files and from corresponding author on request.

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

## Acknowledgements

This research was supported by EPSRC (EP/W024284/1), National Natural Science Foundation of China (22293020, 22293022), National Key R&D Program of China (2021YFE0115800), WBI-MOST (SUB/2021/IND493971/524448), and Royal Society Newton International Fellowship (Grant No. NIF\R1\211458). B. Z. would like to acknowledge CSC-Cambridge scholarship for financial support.

## Author contributions

B.Z. and T.H. conceived the idea of the project. B.Z. performed the theoretical work. B.Z., Zhuo Chen, Y.K., H.M. designed and conducted the experiments, and analysed the data. Q.C., D.L., G.Y. performed the hydrodynamic simulation. Zesheng Chen, B. Z. performed the code and analysis for pore size measurement. B.Z. wrote the draft manuscript. E.A.M., J.C., T.B., P.K.K., L.G.O., J.W.G., B.L.S., and T.H. revised the manuscript. T.H. supervised the project.

## Competing interests

The authors declare no competing interests.
