## [Peer Review File · Nature Communications]

Universal Murray's law for optimised fluid transport in synthetic structuresREVIEWER COMMENTS

Reviewer #1 (Remarks to the Author):

This is an excellent piece of work submitted by Hasan et al, which lays a solid theoretical foundation of Murray's law in synthesizing hierarchically porous materials, with a broad scope for applications involving mass transfer. This work is of great interest because materials following Murray's law are of significant interest due to their unique porous structure and optimized mass transfer ability, which are definitely the important target to pursue both for academia and industry and highly challenging.

The authors firstly propose a Universal Murray's law applicable to a wide range of hierarchical structures, shapes and generalised transfer processes. Then they validate their proposed law through sufficient experiments and further show how simple structural optimisation guided by the proposed theory yields a significant dynamic performance improvement in GOA-based gas sensors. All the experiments have been very carefully designed and performed. The paper of high quality is well organized and well written, easy to read with important support information. The data are convincing and strongly support the discussion and the conclusion. Overall, this is a very interesting and TOP level work, could stimulate a new research field among hierarchical porous materials. I highly recommend acceptance of this present manuscript in Nature Communications for its importance in a rapidly evolving field of high current interest.

I have only one minor concern, Figure 2 provides the average pore size of unidirectionally freeze-cast GOA frozen at different temperatures (e) and average layer height of bidirectional freeze-cast GOA at different temperatures (j). But the SEM images only show Top-view SEM images of unidirectional freeze-cast GOA at -20 °C (b), -40 °C (c), and -70 °C (d) and bidirectional freeze-cast GOA at -10 °C (b), -30 °C (c), and -50 °C (d). What about the Top-view SEM images of unidirectional and bidirectional freeze-cast GOA at other temperatures? Please provides them in the supplementary information.

Reviewer #2 (Remarks to the Author):

In this research work, Zhou et al. proposed a universal Murray's law for optimised fluid transport in synthetic structures. This work gives an important insight into Murray's law by expanding its general form and verifying with the graphene oxide aerogel. This paper can be published in Nature Communications after some suitable revision. Here are my comments.

1. In page 4, the authors pointed out that "The cross-sectional area of a channel can be written as $A=k_1x^\alpha$. where k_1 is the linear coefficient, x is a size variable of the channel, and α is the exponent." This result is an important mathematics foundation for equation derivation and should be appropriately cited in manuscript, as well as in Supporting Information. I am concerned that is this equation suitable for any channels with irregular geometrical shape?

2. If the channel size diminishes to nano or subnano scale, is the equation ($A=k_1x^\alpha$) capable to be used? This is also essential.

3. In authors' deduction, the most general form of Murray's law can be expressed as the function of two exponents (α and β) and size variable of the channel (x). Essentially, α and β are also related to the channel shape. Is there some constraint relationship between these two exponents or size variable of the channel? Actually, it would be better to further describe or define α and β in the paper.

4. The authors discussed the deduction in range of Hagen-Poiseuille flow, which is a kind of laminar flow. Although other kind of Couette flow is not suitable for studying the Murray's law, what about the turbulence? Is this universal Murray' law suitable for turbulence?

5. How did the authors realize the temperature gradient in the vertical direction to samples on a horizontal freezing platform of copper? The corresponding methods part did not show this specific method. This is essential to fabricating multiscale graphene. On the other hand, the macroscopic photograph of vertically porous GOA should be put in Figure 2a, for better showing the porous levels of GOA.

6. The authors used 2-butanol, hexane, ethanol, toluene to verify but these organics do not have obvious differences in fluid properties. What about the glycerol? Experiments about glycerol with high viscosity are encouraged to perform, and corresponding discussions are needed to

supplement.

7. In Figure 4c, is the lattice fringe of SnO₂ or GO? Corresponding XRD patterns of the materials should be added in Supporting Information.

8. For GOA-based gas sensor, I am curious about that how to confirm the response time and recovery time from Figure 4g?

9. Some minor issues. The format in Supporting Information should be modified, for instance, annotations need to be aligned at both ends.

Reviewer #3 (Remarks to the Author):

Noteworthy results

The manuscript consists of (1) a theoretical discussion of Murrays's law and variants, and (2) an experimental investigation of laminar flow in a graphene oxide aerogels (GOAs) with controlled branching pores.

With regards to (1) the authors consider a system where,

* The work of transport P_f is a power law wrt conduit width dimension.

* The work of transport P_f is a power law wrt pressure drop per unit length.

* The conduit volume V is a power law wrt conduit width dimension.

* While not explicitly stated, a power loss proportional to V is assumed.

Within this class of systems an generalized to Murray's law is formulated. Many such systems are identified in the discussion.

With regards to (2), the authors consider an isochoric constraint $V=V_0$. Under this constraint, minimization of work can be replaced by minimization of hydrodynamic resistance. The authors then demonstrate using branching sheets of GOA, and branching ducts of GOA, respectively, that Murray's law holds for laminar flow in these structures. The new observation here is that it works for this particular GOA material.

Significance to field and adjacent fields

There are many previous investigations into the extension of Murray's law to various biological and synthetic systems. The theoretical discussion herein organizes a subclass of these systems, namely the ones mentioned above, into a unified theoretical framework. My judgement, however, is that the contribution does not contribute to novel physics in the sense that it could spark innovation or new avenues of research.

The fabrication of well-controlled GOA structures, and the demonstration that Murray's law holds for laminar flow in these materials is not by itself of interest to a general readership. Moreover, it only covers the already most well-understood aspect of Murray's law, namely it s application to laminar flow.

My judgement is that the contribution to adjacent fields is minor.

Support of conclusions

It would be impossible to investigate the whole range of applications to the generalized law. Stil, it should be said that a very limited range of applicable systems were investigated in this work.

Flaws in the analysis

The theoretical analysis, which is placed in supporting information, was presented by expanding the original Murray's law step by step. This is not normal in theoretical development. A better, and more conventional, approach would be to carefully define the broadest possible assumptions for the class of systems under consideration, and derive the general governing equations. Then, it can be demonstrated with ease that the original Murray's law emerges as a special case.

The same style of incremental discussion is employed in the main text as well, leading to an uneconomical writing style.

Methodology

Well executed.

Reproducibility

There is enough detail for reproducing the results.

Additional comments

There are some grammatical errors.

It is sometimes stated, in the abstract for instance, that materials that follow Murray's law has optimal mass transfer. This is misleading, since Murrays' law emerging from a balance between mass transport power and the cost of the medium. Or, you could be referring to optimal mass transfer under a volume constraint?

In the main text, it is never explicitly stated the resistance R only has a minimum for the Murray material when there is a volume constraint on the material.

It is not clear in "Expanding Murray's law for..." why and how noncircular channels can be parameterized with only one parameter. Explain.

RESPONSE TO REVIEWERS' COMMENTS

We thank the reviewers for their constructive comments and suggestions on our manuscript. Accordingly, we have carried out additional experiments and added more results to address each of the reviewers' comments in detail. Their original comments are marked in **BLUE** here. The changes are marked in **RED** in the revised manuscript and this response.

Reviewer #1

This is an excellent piece of work submitted by Hasan et al, which lays a solid theoretical foundation of Murray's law in synthesizing hierarchically porous materials, with a broad scope for applications involving mass transfer. This work is of great interest because materials following Murray's law are of significant interest due to their unique porous structure and optimized mass transfer ability, which are definitely the important target to pursue both for academia and industry and highly challenging.

The authors firstly propose a Universal Murray's law applicable to a wide range of hierarchical structures, shapes and generalised transfer processes. Then they validate their proposed law through sufficient experiments and further show how simple structural optimisation guided by the proposed theory yields a significant dynamic performance improvement in GOA-based gas sensors. All the experiments have been very carefully designed and performed. The paper of high quality is well organized and well written, easy to read with important support information. The data are convincing and strongly support the discussion and the conclusion. Overall, this is a very interesting and TOP level work, could stimulate a new research field among hierarchical porous materials. I highly recommend acceptance of this present manuscript in Nature Communications for its importance in a rapidly evolving field of high current interest.

Reply: Thank you sincerely for the high evaluation and constructive suggestion on our manuscript, which has been instrumental in improving the overall quality of our work.

Comment #1:

I have only one minor concern, Figure 2 provides the average pore size of unidirectionally freeze-cast GOA frozen at different temperatures (e) and average layer height of bidirectional freeze-cast GOA at different temperatures (j). But the SEM images only show Top-view SEM images of unidirectional freeze-cast GOA at -20 °C (b), -40 °C (c), and -70 °C (d) and bidirectional freeze-cast GOA at -10 °C (b), -30 °C (c), and -50 °C (d). What about the Top-view SEM images of unidirectional and bidirectional freeze-cast GOA at other temperatures? Please provides them in the supplementary information.

Reply: Thank you for this suggestion. As per the reviewer's suggestion, we now have added the top-view SEM images of all freeze-cast GOA in Supplementary Figs. 4 and 9, including the aerogels unidirectionally freeze-cast at -10 to -80 °C and bidirectionally freeze-cast at -10 to -50 °C. The figures are reproduced below.

Supplementary Figure 4. Top-view SEM images of unidirectionally freeze-casted GOA frozen at (a) -10 °C, (b) -20 °C, (c) -30 °C, (d) -40 °C, (e) -50 °C, (f) -60 °C, (g) -70 °C, and (h) -80 °C. Scale bars: 100 μm.

Supplementary Figure 9. Top-view SEM image of bidirectionally freeze-casted GOA frozen at (a) $-10\text{ }^{\circ}\text{C}$, (b) $-20\text{ }^{\circ}\text{C}$, (c) $-30\text{ }^{\circ}\text{C}$, (d) $-40\text{ }^{\circ}\text{C}$, and (e) $-50\text{ }^{\circ}\text{C}$. Scale bars: $100\text{ }\mu\text{m}$.

Additionally, we have mentioned referred to these figures in the main text:

(Page 8, Line 9) The vertical pores in GOA frozen at different temperatures are shown in Fig. 2b-d and Supplementary Fig. 4.

(Page 8, Line 41) The aerogel, which is bidirectionally freeze-cast at a relatively high temperature, shows a regular lamellar architecture in the top-view (Fig. 2g Supplementary Fig. 9) and side-view section (Supplementary Fig. 10).

Reviewer #2

In this research work, Zhou et al. proposed a universal Murray's law for optimised fluid transport in synthetic structures. This work gives an important insight into Murray's law by expanding its general form and verifying with the graphene oxide aerogel. This paper can be published in Nature Communications after some suitable revision. Here are my comments.

Reply: Thank you for the careful review and high evaluation of this work. Your insightful comments have played a critical role in the enhancement of our manuscript.

Comment #1:

In page 4, the authors pointed out that "The cross-sectional area of a channel can be written as $A=k_1x^\alpha$. where k_1 is the linear coefficient, x is a size variable of the channel, and α is the exponent." This result is an important mathematics foundation for equation derivation and should be appropriately cited in manuscript, as well as in Supporting Information. I am concerned that is this equation suitable for any channels with irregular geometrical shape?

Reply: Thank you for this comment. **Indeed the formula represents a general expression for channels with any geometrical shape rather than a mathematical result.** Let us consider a very general situation in Universal Murray's law. For a hierarchical channel with an unspecified shape, we start with a proper variable x to describe the size of the channel. We then **assume a power law relationship** between the cross-sectional area A and the size variable x , written as $A = k_1 x^\alpha$. In other words, for any channel with regular or irregular geometrical shape, **we select a proper variable x satisfying $A = k_1 x^\alpha$ to start the deduction.**

This starting point is general for any channel. For most closed 2D shapes, it is obvious when the shapes enlarge or shrink, the section area is proportional to the square of the radius, diameter, perimeter, side lengths, and hydraulic diameter (See Supporting Information: Murray's law in hierarchical tubular network with arbitrary shape). The irregular shape can be divided into infinitesimally small and regular shapes, such as rectangular or triangles, whose area is proportional to the square of the length. In these cases, the above common variables give $\alpha = 2$, which is sufficient for the following deduction.

Our Universal Murray's law based on this assumption even allows other uncommon preconditions following $A = k_1 x^\alpha$. For example, in the deduction of Equ. (7) in the main text, we can directly choose the sectional area as variable, $x = A$. Thus, $A = k_1 x^\alpha$ directly transforms into $A = A$, with k_1 and α being 1. Other uncommon but proper variables, such as angle, may also be used when necessary.

The reviewer's question is important to clarify our approach. Therefore, in response, we now have revised the relevant discussions in the main text and Supporting Information as below:

(Page 4, Line 28) **For a proper size variable of the channel x , if we assume the cross-sectional area has a power law relationship with x , we can write $A = k_1 x^\alpha$, where k_1 is the linear coefficient, and α is the exponent of x in power function A .**

(Supporting Information, Page 7) For a proper size variable x grown with the cross-sectional area, such as radius r for the circular tube or side length for the square section, **assuming** the cross-section area A can generally be written as: $A = k_1 x^\alpha$.

Comment #2:

If the channel size diminishes to nano or subnano scale, is the equation ($A=k_1x^\alpha$) capable to be used? This is also essential.

Reply: Thank you for the question. In short, **YES.** The Universal Murray's law and its derivatives rely on mathematical relations, irrespective of the length units. Therefore, this entire principle, not only $A = k_1 x^\alpha$, applies to **arbitrary length scales**. Specifically, $A = k_1 x^\alpha$ depends on the channel shape and the selection of size variable x , rather than the scale of the channel (e.g. Circles at the nanometre scale and the kilometre scale both follow $A = \pi r^2$).

Concerning the channel scale, using Universal Murray's Law requires particular attention to the applicability of the transfer process rather than the cross-sectional area. For example, in the diffusion process, the molecular diffusion rate in macroscopic columnar channel is proportional to the square of radius according to Fick's first law, $Q \propto r^2$. This leads to Murray's law with an exponent of 2 for r in the main text Equ. (2): $\sum r_1^2 = \sum r_2^2 = \sum r_3^2 = \dots$. However, when the channel dimension reduces to the same order as the mean free path of diffusive molecules (e.g. 60 nm for N₂ gas at standard state), Knudsen diffusion must be considered. When Knudsen diffusion is dominant, the diffusion rate $Q \propto r^3$ for columnar channel, and the resultant expression has an exponent of 2.5 in Equ. (8). When the dimension further diminishes to a sub-nano scale (i.e. lower than twice the Debye length), ionic transfer behaves differently in nanofluidic channels based on the charge and the preconditions based on the above transfer types do not hold. (*Science*, 351, 1395-1396) In conclusion, **the validity and applicability of Universal Murray's law and its derivatives are independent of the channel scale**. However, the user should choose the proper transfer type based on the channel dimension at the microscopic scale to provide the correct precondition.

Comment #3:

In authors' deduction, the most general form of Murray's law can be expressed as the function of two exponents (α and β) and size variable of the channel (x). Essentially, α and β are also related to the channel shape. Is there some constraint relationship between these two exponents or size variable of the channel? Actually, it would be better to further describe or define α and β in the paper.

Reply: Thank you for this insightful question. **We do not establish or find any constraint relationship between α and β in general**. Although they both rely on the shape and the selected variable x , β also depends on the transfer type. Therefore, it is unfeasible to derive a general relationship between α and β for those different transfer types.

In some specific transfer types, α and β can be related. For example, for diffusion or ionic transfer, the exponents of x in Q and A should be equal ($\alpha = \beta$), due to the proportionate relationship of transfer rate and channel, $Q \propto A$:

(Page 5, Line 32) This is consistent with Pouillet's law and Ohm's law for ionic and electronic transfer $Q = \sigma A \cdot \frac{\Delta V}{l}$ (where σ is the conductivity and ΔV is the potential difference) and Fick's law for diffusion $Q = DA \cdot \frac{\Delta C}{l}$, (where D is the diffusion coefficient and ΔC is the concentration difference).

The relationship between exponents and variable x comes from the assumptions of the power law, similar to the discussion in the first question. For variable x , we assume the power relationship exists between the area A and x , and the transfer rate Q and x , then we can write down the first two equations, $A = k_1 x^\alpha$ and $Q = k_2 x^\beta \cdot \frac{\Delta P}{l}$.

Beyond the above prerequisites, we do not establish other redundant constraint

relationships between the exponents and x . Thus, the selection of x can be quite arbitrary and flexible in practice, aiming to make the following actual calculations as convenient as possible.

Accordingly, we have already updated Page 4 Line 28 in our revised manuscript (Page 4 in this response letter). Additionally, we have clarified some descriptions for α and β in our revised manuscript:

(Page 4, Line 32) Supposing a potential ΔP drives a mass transfer process in the network, **and assuming the generalised mass flow rate Q also has a power relationship with x , written as $Q = k_2 x^\beta \cdot \frac{\Delta P}{l}$** , where k_2 is the linear coefficient, l is the channel length of a section, and β is the exponent **of x in power function Q** .

(Supporting Information, Page 7) **Assuming the transfer rate Q varies as a power of x** , it can be expressed as: $Q = k_2 x^\beta \cdot \frac{\Delta P}{l}$

Comment #4:

The authors discussed the deduction in range of Hagen-Poiseuille flow, which is a kind of laminar flow. Although other kind of Couette flow is not suitable for studying the Murray's law, what about the turbulence? Is this universal Murray' law suitable for turbulence?

Reply: Thank you for this insightful question. In this work, **we did use Universal Murray's law to optimise turbulent flow in theory** and obtain consistent results with published works, as discussed in the original manuscript and Supporting Information: Several derivations of Universal Murray's law:

(Page 5, Line 10) Equation (6) can be used to readily optimise the turbulent flow in rough pipes ($\sum r_i^{7/3} = \dots = \sum r_i^{7/3}$), turbulent flow in smooth pipes ($\sum r_i^{17/7} = \dots = \sum r_i^{17/7}$), and laminar flow of non-Newtonian liquids following power-law rheology ($\sum r_i^3 = \dots = \sum r_i^3$), consistent with the reported results (for details, see Supporting Information: Several derivations of Universal Murray's law).

The experimental verification of turbulent flow is impractical for our hierarchical GOA material system. Single-phase pipe flow tends to be turbulent when the Reynolds number is larger than 4000, $Re = \frac{uL}{\nu} > 4000$, where u is the flow rate, ν is the kinematic viscosity of the fluid, and L is the characteristic linear dimension. (*AIChE J.* 5, 433-435) Considering the water flow in the vertically-porous aerogel freeze-cast at -20 °C (20.6 μm pore diameter), the flow rate needs to be larger than 196 m s^{-1} to achieve turbulent flow. In a typical straight pipe used for verification and gas sensing in this paper (Fig. R1), with a diameter of 3.46 mm, this value corresponds to a volumetric flow rate of $1.11 \times 10^5 \text{ mL min}^{-1}$. The volumetric flow rate needs to be larger than $1.62 \times 10^6 \text{ mL min}^{-1}$ for air flow. The aerogel freeze-cast at lower temperatures with smaller pores requires an even higher flow rate. **Our aerogel obviously cannot handle such a high flow rate.** Note that the Reynolds numbers in all our experiments are at least four orders of magnitude smaller than the critical

value of laminar flow, 2000.

Figure R1. Schematic demonstration of typical straight pipe with a diameter of 3.46 mm, filling with vertically porous aerogel of 20.6 μm pores.

Comment #5:

How did the authors realize the temperature gradient in the vertical direction to samples on a horizontal freezing platform of copper? The corresponding methods part did not show this specific method. This is essential to fabricating multiscale graphene. On the other hand, the macroscopic photograph of vertically porous GOA should be put in Figure 2a, for better showing the porous levels of GOA.

Reply: Thank you for these important suggestions. In response to the first question, we measured the temperature change of GO ink during freezing, as shown in Supplementary Fig. 3 below. The temperature gradient is achieved by **placing the GO ink at room temperature on the cold copper platform**. The copper platform is precooled and maintained at a certain temperature. When the GO hydrogel is put on the cold platform, the bottom of the sample in contact with the copper cools down rapidly, while the temperature change at the top is relatively slow. In this way, a vertical temperature gradient is established, as shown in the Supplementary Fig. 3a (also reproduced below). The temperature at the Bottom is always lower than that at the Middle and the Top during the freezing process and approaches the freezing point earlier. This results in ice crystal growth from the bottom to the top. Similarly, in the bidirectional freeze-casting process, the Top position is colder than the Side, although they are at the same height (Supplementary Fig. 3b). This shows that a thermally-isolating PDMS wedge creates another horizontal temperature gradient in bidirectional freeze-casting.

Supplementary Figure 3. Temperature gradient in directional freezing. (a) Temperature changes at different heights during the unidirectional freezing of GO ink at $-20\text{ }^{\circ}\text{C}$. The thermometer probe is inserted at vertical axis along the centre of a 1 cm cube, at the height of 2.5 mm (Bottom), 5 mm (Middle), and 7.5 mm (Top). (b) Temperature changes during the bidirectional freezing of GO ink at $-20\text{ }^{\circ}\text{C}$. The thermometer probe is inserted into a 1 cm cube. The Top and Side are at the same height, and the Top and Bottom are at the same horizontal position.

In response to the reviewer’s comment, we now have added the discussion in the main text and revised the Methods:

(Page 6, Line 42) As shown in Fig. 2a, we use the unidirectional freeze-casting method to prepare the vertically-porous GOA. This method applies a temperature gradient in the vertical direction to samples on a horizontal freezing platform of copper. **Temperature changes at different heights during the unidirectional freeze-casting demonstrate the vertical temperature gradient and the bottom-to-top ice growth (Supplementary Fig. 3a).**

(Page 8, Line 35) When a polydimethylsiloxane (PDMS) wedge with a certain angle is inserted between the cooling stage and the sample being freeze-cast, an additional horizontal temperature difference (Fig. 2f) is applied in the deposited GO due to the low thermal conductivity of PDMS.[41, 43] **As shown in Supplementary Fig. 3b, the sequence of cooling and freezing at different positions reveals the temperature gradients in two directions during the bidirectional freezing.**

(Page 14, Line 33, Methods) The freeze-casting method utilises a custom-built

freezing device **with a copper freezing platform as cold source**. In unidirectional freeze-casting, the samples **at room temperature** are directly placed on **the precooled** platform. **The platform is precisely maintained at a specific temperature for 30 min**. As for bidirectional freeze-casting, the samples are placed on a **precooled** 30°PDMS wedge on the platform for 30 min.

As for the macroscopic photograph of GOA, we have taken a photo of **GOA standing on the fragile bristles to show the lightness and high porosity** (Supplementary Fig. 2). This vertically-porous aerogel is unidirectionally frozen at $-20\text{ }^{\circ}\text{C}$. We put this photo into Supporting Information instead of Fig. 2 to keep Figures in the main text informative and tidy. Correspondingly we mentioned in the main text:

(Page 6, Line 36) The density of the as-prepared GOA (see Methods) is **low to 25 mg cm^{-3}** (Supplementary Fig. 2).

Supplementary Figure 2. GOA on the bristles of green foxtail.

We have compared macroscopic photos of vertically-porous GOA frozen at different temperatures and GOA frozen by liquid nitrogen, as shown in Fig. R2 below. In photographs, their surfaces do not exhibit significant differences in appearance to indicate the differences in their porous structures.

Figure R2. Macroscopic images of GOA samples GOA. (a-c) Photos of vertically-porous GOA freeze cast at (a) $-20\text{ }^{\circ}\text{C}$, (b) $-40\text{ }^{\circ}\text{C}$, and (c) $-70\text{ }^{\circ}\text{C}$. (d) Photo of GOA frozen by liquid nitrogen.

Additionally, we have taken top-view microscopic photos of GOA (Fig. R3a-c below). Compared with corresponding SEM images (Fig. R3d-f), microscopic photos also show a consistently ordered porous structure. The pore size gradually decreases as the freezing temperature reduces from $-20\text{ }^{\circ}\text{C}$ to $-70\text{ }^{\circ}\text{C}$. However, only a thin section in the 3D structure of aerogel is clearly observed. The other parts appear blurry due to limited focus and the uneven nature of the broken GOA surface. We therefore conclude that the **clearer SEM images are sufficient to demonstrate GOA microstructure at different levels.**

Figure R3. Top-view images of vertically porous GOA. (a-c) Microscopic images of GOA unidirectionally frozen at (a) $-20\text{ }^{\circ}\text{C}$, (b) $-40\text{ }^{\circ}\text{C}$, and (c) $-70\text{ }^{\circ}\text{C}$. (d-f) Corresponding SEM images of GOA unidirectionally frozen at (d) $-20\text{ }^{\circ}\text{C}$, (e) $-40\text{ }^{\circ}\text{C}$, and (f) $-70\text{ }^{\circ}\text{C}$. Scale bars $100\text{ }\mu\text{m}$.

Comment #6:

The authors used 2-butanol, hexane, ethanol, toluene to verify but these organics do not have obvious differences in fluid properties. What about the glycerol? Experiments about glycerol with high viscosity are encouraged to perform, and corresponding discussions are needed to supplement.

Reply: Thank you for this suggestion. **We have found that highly viscous fluids like glycerol (viscosity $\eta = 150\text{ cPa s}$) are impractical for our experiments.** Because of high viscosity, the large drag force often damages aerogel's microstructure. In the experiment, we always observed black aerogel fragments flowing out with glycerol. Additionally, pipes were almost blocked after glycerol flow. These results imply that the ordered pores in the GOA sample are destroyed, preventing us from obtaining the actual resistance of hierarchical pores for highly viscous liquid.

We believe **our current results are sufficient to support the universality of the proposed law**, which has been verified in 6 polar or nonpolar fluids with diverse viscosities: water, air, 2-butanol, hexane, ethanol, and toluene. Their viscosity ranges from 0.0181 cPa s (air) to 3.95 cPa s (2-butanol), across 2 orders of magnitude. Furthermore, **the fluid viscosity does not affect the optimisation result in theory**, regardless of its value. Considering laminar flow in the deduction of Universal Murray's law, the viscosity term will be eliminated as part of k_2 term, as shown in the following Supporting Information:

(Supporting Information, Page 7)

$$Q = k_2 x^\beta \cdot \frac{\Delta P}{l} \quad (S.19)$$

where k_2 is the linear coefficient and β is the exponent of x . For example, Hagen-Poiseuille's law gives $Q = \frac{\pi r^4}{8\eta l} \Delta p$ for laminar flow in tube, thus, here $x = r$, $k_2 = \frac{\pi}{8\eta l}$, and $\beta = 4$.

.....

setting constant $k_3 = -\frac{\beta}{\lambda k_1 k_2 \alpha}$, we have:

$$n_1^2 x_1^{\alpha+\beta} = n_2^2 x_2^{\alpha+\beta} = \dots = n_i^2 x_i^{\alpha+\beta} = k_3 \quad (S.23)$$

Comment #7:

In Figure 4c, is the lattice fringe of SnO₂ or GO? Corresponding XRD patterns of the materials should be added in Supporting Information.

Reply: Thank you for this constructive suggestion. **In short, SnO₂ quantum dots.** In response to your question, we took the TEM images of pure SnO₂ QDs and graphene oxide, shown in Fig. R4. Compared to SnO₂ QD-decorated GO (Fig. 4b-d as below), TEM image of **pure SnO₂ QDs (Fig. R4b) also shows lattice fringes** of 3.3 Å and 2.5 Å, corresponding to (110) and (001) of SnO₂, respectively (*J. Mater. Sci.*, 43, 5291-5299.). The pure GO only have curved and immeasurable strips, as shown in Fig. R4c-e. Therefore, we suggest that the lattice fringes in Fig. 4b-c are of SnO₂ QDs.

In response to your enquiry, we have also added an enlarged TEM image in Fig. 4d to clearly show the lattice fringes.

Figure R4. TEM images of (a-b) SnO₂ quantum dots, and (c-e) graphene oxide. Scale bars (a, d) 10 nm, (b, e) 1 nm, and (c) 10 μm.

Figure 4b-d. Optimising tubular GOA-based gas sensor by Murray's law. (b-d) TEM images of SnO₂ QD-decorated GO ink. Scale bars: (b) 100 nm, (c) 10 nm, and (d) 2 nm.

We have also added the XRD pattern of SnO₂ QD-decorated GO and SnO₂ QDs in Supplementary Fig. 19. Correspondingly, we have now discussed the result in the main text:

(Page 12, Line 26) X-ray diffraction (XRD) patterns in Supplementary Fig. 19 of SnO₂ QD-decorated GO and SnO₂ QDs can be attributed to the tetragonal phase of SnO₂ (JCPDS card no. 41-1445). SnO₂ QD-decorated GO shows an additional characteristic diffraction peak of GO (001).[45]

(Page 15, Line 3, Methods) The SnO₂ QDs are prepared in the same way without

the GO dispersion.

(Page 15, Line 7, Methods) XRD patterns were measured from 5° to 90° by a Bruker D8 Advance powder X-ray diffractometer with CuK α (1.54 Å) radiation.

Supplementary Figure 19. XRD patterns of SnO₂ QD-decorated GO and SnO₂ QDs.

Comment #8:

For GOA-based gas sensor, I am curious about that how to confirm the response time and recovery time from Figure 4g?

Response: Thank you for this important question. We draw a response schematic (Fig. R5) to show the definition of response and recovery time. The response time τ_{res} for gas sensor is the time required to reach a stable output after the sensor is exposed to the target gas (gas ON), normally regarded as reaching 90% of the maximum response. The recovery time τ_{rec} is the time required for returning to 10% of the maximum response after the sensor is exposed to clean air (gas OFF). (*Chem. Soc. Rev.*, 49(6), 1756-1789) In practice, we first find the highest point in the response curve, which is normally close to the end of target gas exposure. Then, we mark 90% response in the response curve and 10% in the recovery curve and calculate the corresponding τ_{res} and τ_{rec} from the response curve:

(Page 16, Line 21, Methods) The response time is defined as the time to reach 90% of the maximum response after the exposure to target gas. The recovery time is defined as the time spent for the response to return to 10% above the baseline after the removal of the target gas.

Figure R5. Diagram illustrating the definition of the response time and recovery time.

In response to this question, we have also added gas sensing test to 1 ppm ammonia and formaldehyde and extended the exposure time to 1 hr. These additional experiments further confirm the improvement of response and recovery time of the gas sensor following Murray’s law. As shown in Fig. 4h and Supplementary Fig. 20 below, in all three types of sensing gases, the response and recovery times of the gas sensors following Murray’s law are shorter than the hierarchically porous counterparts. The reduction in response and recovery times is close to the simulated decrease in flow resistance (Fig. 4i).

We have also modified the corresponding discussions in the main text:

(Page 12, Line 39) This simple shape adjustment based on Murray’s law shortens the response times τ_{res} and recovery times τ_{rec} by from 8.6% to up to 18.2% in the gas sensing of 1 ppm NO_2 , NH_3 , and CH_2O (Fig. 4h and Supplementary Fig. 20).

Figure 4h. Response and recovery time of SnO_2 QDs-doped GOA in the hierarchical straight pipe and the pipe optimised by Murray’s law to 1 ppm nitrogen dioxide, ammonia, and formaldehyde.

Supplementary Figure 20. Response curves of hierarchical SnO_2 QD-doped GOA in the straight pipe and optimised by Murray's law towards (a) 1 ppm nitrogen dioxide, (b) 1 ppm ammonia, and (c) 1 ppm formaldehyde.

Comment #9:

Some minor issues. The format in Supporting Information should be modified, for instance, annotations need to be aligned at both ends.

Reply: Thank you for the careful review. We have carefully checked and made corresponding format revisions in accordance with your suggestion. We also checked and revised other formatting issues in the Supporting Information document.

Reviewer #3

Noteworthy results

The manuscript consists of (1) a theoretical discussion of Murrays's law and variants, and (2) an experimental investigation of laminar flow in a graphene oxide aerogels (GOAs) with controlled branching pores.

With regards to (1) the authors consider a system where,

- * The work of transport P_f is a power law wrt conduit width dimension.
- * The work of transport P_f is a power law wrt pressure drop per unit length.
- * The conduit volume V is a power law wrt conduit width dimension.
- * While not explicitly stated, a power loss proportional to V is assumed.

Within this class of systems an generalized to Murray's law is formulated. Many such systems are identified in the discussion.

With regards to (2), the authors consider an isochoric constraint $V=V_0$. Under this constraint, minimization of work can be replaced by minimization of hydrodynamic resistance. The authors then demonstrate using branching sheets of GOA, and branching ducts of GOA, respectively, that Murray's law holds for laminar flow in these structures. The new observation here is that it works for this particular GOA material.

Reply: Thank you for the summary. We greatly appreciate your review of our manuscript. However, this summary has some **significant misunderstandings** about our work.

(1) In the first part about theory, **none of the four conditions listed above are considered** in our theoretical development of Universal Murray's law and its derivatives. They are actually what **Cecil D. Murray used in his original publication of Murray's law in 1926** (*Proc. Natl. Acad. Sci. U. S. A.*, 12, 207-214). We discuss this pioneering work at the beginning of the manuscript (Page 2, Line 28) and briefly introduce his deduction in Supporting Information: The initial deduction of Murray's law. **Our theory starts with a set of totally different preconditions.**

In response to the 1st and 2nd preconditions listed above, we consider minimising **flow resistance R instead of work of transport P_f** in our theory. This is one of the critical modifications to Murray's initial work. As discussed in our original manuscript, this change allows us to investigate the general transfer process instead of only laminar flow:

(Page 4, Line 2) The minimisation of R is equivalent to the maximisation of efficiency, where ΔP is optimally utilised to drive the flow. Intuitively, the deduction of minimising resistance generates the identical cubic form of Murray's law for laminar flow and the square form for diffusion and ionic transport (for details, see Supporting Information: Deducing Murray's law by minimising resistance). Compared to the original deduction by investigating power cost, **minimising resistance offers a more versatile approach allowing extension of the law to other transfer types** like diffusion, where quantifying energy consumption is challenging.

Regarding the 3rd precondition listed above, as shown in the main text below, we assume that the **channel section area A varies as the power law of size variable x** . Although this assumption can also lead to the power law relationship of the volume V and variable x , x is not limited to just the width dimension. It also allows other uncommon variables, such as the sectional area A itself in Equ. (7).

(Page 4, Line 28) **For a proper size variable of the channel x , assuming the cross-sectional area has a power law relationship with it, and thus can be written as $A = k_1 x^\alpha$, where k_1 is the linear coefficient, and α is the exponent of x in power function A .**

The 4th listed assumption is not mentioned in our theory at all. Our theory does not involve power loss or transfer work. Additionally, as a widely used assumption in biomechanics, a metabolism power loss proportional to volume cannot be directly applied in synthetic materials. As an alternative, the total volume of the hierarchical structure is fixed for porous materials.

(Page 3, Line 1) Later, it was proposed that vascular systems can be optimised within **the confines of a given total volume** instead of considering tissue metabolism by following various forms of Murray's law described earlier.[24, 31, 32] **This alternative approach enables the application of Murray's law in optimising the mass transfer of synthetic porous materials**, as the 'mass transfer performance' of materials should be considered as an intensive property, referring to the mass transport capacity per unit volume.

(2) With regard to the second experimental part, **the purpose of the our experiments is to verify Universal Murray's law, not just to demonstrate Murray's law in a particular GOA materials**. As discussed in the main text below, we first predict the optimal hierarchically planar structure by Universal Murray's law, which has never been discussed before. Then, we experimentally verify it in lamellar GOA. **This prediction-verification process** is the general paradigm for testing the new theory. Thus, we argue that **the experiments in planar structure can be solid evidence for the validity of Universal Murray's law**. The

following examinations in tubular structure can further prove our theory as a well-studied model. In the experiments, GOA is just an appropriate porous material we chose to construct the proposed hierarchical structure. Although we used quite innovative method to prepare the unique structure in GOA, the material itself is not and should not be the primary focus of this work.

(Page 11, Line 12) The examination of the planar structure for optimal laminar flow is convincing evidence for the establishment of Universal Murray's law. **For the first time, we expand Murray's law into a hierarchically planar structure and experimentally confirm it.** Both the experiments and simulation demonstrate that the branching lamellar GOA structures achieve minimised resistance for laminar flow when following the corresponding expression of Universal Murray's law.

(Page 12, Line 2) As a more classic and frequently discussed model[23, 24, 28], these results of tubular structure **further validate the Universal Murray's law in materials** by both experimentation and simulation.

Besides, the isochoric constraint $\sum V = V_0$ and the minimisation of resistance is consistently employed in our theoretical part, not only in the experimental part.

Significance to field and adjacent fields

There are many previous investigations into the extension of Murray's law to various biological and synthetic systems. The theoretical discussion herein organizes a subclass of these systems, namely the ones mentioned above, into a unified theoretical framework. My judgement, however, is that the contribution does not contribute to novel physics in the sense that it could spark innovation or new avenues of research.

The fabrication of well-controlled GOA structures, and the demonstration that Murray's law holds for laminar flow in these materials is not by itself of interest to a general readership. Moreover, it only covers the already most well-understood aspect of Murray's law, namely its application to laminar flow.

My judgement is that the contribution to adjacent fields is minor.

Reply: Thank you for the critical comment on our work. We respectfully disagree. As noted by the comment, **Murray's law has been well-studied in biomechanics and hydrodynamics** in the last century. In biomechanics, numerous vascular theoretical models have been developed based on Murray's law and have been put forward to describe the biological networks (*Bull. math. biophys.* 16, 59-74; *Bull. math. biophys.* 17, 219-227; *Phys. Med. Biol.*, 44, 2929-2945). From plants' xylem (*Nat.*, 421, 939-942) to animals' coronary artery (*J. Gen. Physiol.*, 74, 537-548) and capillary (*Am. J. Physiol.*, 245, H1031-1038), the extensive observations in organisms validated the Murray's law and related models. In hydrodynamics, Murray's law has been expanded into a broader range of situations, including non-Newtonian fluid, yield-stress fluid (*J. Fluid Mech.*, 967, A6), turbulent flow (*Phys. A: Stat. Mech.*, 393, 527-534), and rectangular pipes (*Proceedings of the 3rd European Conference on Microfluidics*. Paper: μ FLU12-235.). The theory has also

been applied in optimising microfluidics (*Lab on a Chip*, 6, 447-454) and micromixers (*Int. J. Heat Mass Transf.*, 141, 346-352). **Since 2017, Murray's law has also attracted great interest from material science.**(*Nat. Comm.*, 8, 14921) Significant recent efforts have been devoted to synthesising Murray materials, defined as the hierarchically porous materials following Murray's law.(*Adv. Mater.*, 34, 2200653; *Carbon*, 150, 21-26; *J. Energy Chem.*, 68, 624-636) However, to the best of our knowledge, **none of the previous research developed a systematic theory regarding a generalised channel shape and transfer type, crucial for synthetic porous materials.**

Our work **first expands Murray's law into arbitrary hierarchical structures, shapes, and transfer processes.** Targeting application of synthetic porous materials for the first time ever since Murray published his paper close to 100 years ago, our **Universal Murray's law and its associated derivatives lay a solid theoretical foundation for the rapid development of Murray materials towards practical applications.**

Besides, in response to the reviewer's concern about the subclasses, Murray's law includes two optimisation aspects: the channel size and numbers (*Proc. Natl. Acad. Sci. U. S. A.*, 12, 207-214), and the position and angle of branching point (*J. Gen. Physiol.*, 9, 835-841). Based on the characteristics of porous material synthesis, our work primarily discussed the optimisation of channel size and branching numbers. The pore size and the ratio of pores at different levels are relatively practical to be controlled in synthesis (*Nat. Commun.*, 8, 14921). Beyond the scope of this paper, we are working on the second optimisation aspect, the branching angle and position, which is more difficult for synthesis and less meaningful for porous materials.

For the contribution of the experimental part, **our work first verified the transfer superiority of obeying Murray's law and Universal Murray's law in porous materials.** Previous studies only compared the porous materials with different levels of hierarchy. (*Nat. Comm.*, 8, 14921; *Inorg. Chem. Front.*, 5, 2829-2835; *J. Energy Chem.*, 68, 624-636) These works only demonstrate the improvement of additional hierarchy levels, instead of Murray materials' superiority. Our verifications compare porous GOA with the same hierarchy but different structures instead. Besides, the concise demonstration of gas sensing also effectively illustrates the transfer superiority of Murray materials.

In response to the comment about the materials, **GOA is just used as an appropriate porous material, chosen to construct the proposed optimal, porous structure for the verification and demonstration of our hypothesis.** Additionally, laminar flow is the most well-known and widely-recognised flow type for Murray's law. It is also the most practical and convenient for verification. Despite only experimenting with laminar flow, our work is sufficient to verify the proposed Universal Murray's law, especially the prediction-examination process in the hierarchically planar structure. We also tried various fluids with considerably different viscosities and polarities. This work provides a very comprehensive system, from the theory to the experimental verification and the practical application. Out of the scope of this paper, we are continuing to experiment with the other common transfer types in the future, such as molecular diffusion, Knudsen diffusion, and ionic transfer.

In conclusion, compared to all papers on Murray's law published since the original work

in 1926, both theoretical and experimental sections of our research contain significantly new information, **contribute to materials science, and will certainly accelerate the synthesis of porous materials following Murray's law. We are confident that our work will attract significant interest** from a wide-ranging readership and qualifies for publication in this journal.

Support of conclusions

It would be impossible to investigate the whole range of applications to the generalized law. Still, it should be said that a very limited range of applicable systems were investigated in this work.

Reply: Thank you for the criticism. We reiterate that **the current examination and demonstration of laminar flow achieve the purposes** of verifying the Universal Murray's law and showing the practical value of Murray materials. Laminar flow is the most common flow type for Murray's law and the most straightforward to examine. We argue that our theoretical derivation, strongly supported by flow experiments of different types of fluids and gas sensing application in structured GOA, represents sufficient direct evidence to convince the community of the validity of our Universal Murray's law. In particular, we show that a simple structural modification guided by our theory significantly enhances the gas sensor's performance, highlighting the process and outcome of optimisation based on the law. Future, unspecified experiments are well beyond the scope of our manuscript.

Flaws in the analysis

The theoretical analysis, which is placed in supporting information, was presented by expanding the original Murray's law step by step. This is not normal in theoretical development. A better, and more conventional, approach would be to carefully define the broadest possible assumptions for the class of systems under consideration, and derive the general governing equations. Then, it can be demonstrated with ease that the original Murray's law emerges as a special case.

The same style of incremental discussion is employed in the main text as well, leading to an uneconomical writing style.

Reply: Thank you for the kind suggestion. Although the proposed writing style could perhaps be better suited for a pure mathematics journal, **our current manuscript is primarily written with chemists, material scientists, and engineers as the target audience.** We note that our work is not an entirely new theory created from scratch. It is an extension based on an existing, well-studied, and well-known law in biomechanics and hydrodynamics. However, **most material scientists, engineers, and chemists, the majority of potential readership of our paper, are not very familiar with Murray's law.** In our original submission, we start the main text by introducing the original Murray's law and then expand it into Universal Murray's law, step by step. This approach helps most readers learn the background of Murray's law, follow our derivation, and understand our theory. Therefore, considering our theory's origin and most readers' background

knowledge, we have decided to keep the current structure.

Methodology

Well executed.

Reproducibility

There is enough detail for reproducing the results.

Reply: Thank you for the acknowledgement of our experiment details.

Additional comments

There are some grammatical errors.

It is sometimes stated, in the abstract for instance, that materials that follow Murray's law has optimal mass transfer. This is misleading, since Murrays' law emerging from a balance between mass transport power and the cost of the medium. Or, you could be referring to optimal mass transfer under a volume constraint?

In the main text, it is never explicitly stated the resistance R only has a minimum for the Murray material when there is a volume constraint on the material.

It is not clear in "Expanding Murray's law for..." why and how noncircular channels can be parameterized with only one parameter. Explain.

Reply: Thank you for the comments.

We carefully checked the main text and Supporting Information and made grammatical revisions.

In response to your second comment about the optimal mass transfer of the materials, we have discussed relevant points in the original manuscript as below (Page 3, Line 1). In short, the **mass transfer performance is an intrinsic and intensive property for porous materials**, like the specific surface area. The internal structure of each material determines its mass transfer performance. It should not change with the volume or the mass of materials. When this paper talks about mass transfer performance, it refers to **mass transport capacity per unit volume**. Unlike biomechanics, optimising porous materials utilises a fixed volume constraint instead of the medium cost. Thus, **the optimal mass transfer or minimised resistance under a volume constraint refers to the optimal mass transfer performance**. Accordingly, we have added a sentence to further explain, and changed "mass transfer" to "mass transfer performance" to avoid misunderstanding.

(Page 3, Line 1) Later, it was proposed that vascular systems can be optimised within the confines of a given total volume instead of considering tissue metabolism by following various forms of Murray's law described earlier.[24, 31, 32] This alternative approach enables the application of Murray's law in optimising the mass

transfer of synthetic porous materials, **as the ‘mass transfer performance’ of materials should be considered as an intensive property, referring to the mass transport capacity per unit volume. The optimisation of porous materials under the volume constraint essentially corresponds to the optimal mass transfer performance.**

(Page 2, Line 21) ...and experimentally confirm optimal mass transfer **performance** for laminar flow using a range of fluids.

(Page 9, Line 21) Therefore, the verification of optimal mass transfer **performance** in Murray materials requires comparison between samples following or deviating from the law but with the same level of hierarchy.

In response to the third additional comment, the volume constraint has been mentioned repeatedly in the original manuscript:

(Page 9, Line 42) We also prepare other hierarchically planar pipes that deviate from Murray’s law and obey the conservation $\sum h_1^x = \sum h_2^x = \sum h_3^x$ for exponent values of $x = 1, 1.5, 2.5, 3$, **with the same channel length, width, and total volume.**

(Page 11, Line 34) Consequently, with a **constrained total volume**, Murray’s law can also be described as a principle that appropriately distributes a larger volume into the high-level sections of smaller channel size and higher resistance, such that it balances the resistance of different sections to minimise the total resistance.

(Page 15, Line 32, Methods) The hierarchically planar structures are designed based on the corresponding equations, **under the constraints of a total volume of 1300 mm³**, a pipe width of 10 mm, and a pipe length of 30 mm. Similarly, the hierarchically tubular structures are designed under the restrictions of **a total volume of 90π** and a fixed length of 30 mm.

(Figure 3 caption) (a-b) The experimental flow resistance of (a) water and (b) air in hierarchical lamellar GOA with **a fixed total volume** and corresponding simulated resistance in scaled-down models.

Accordingly, we have modified some sentences to clarify your concern:

(Page 11, Line 14) Both the experiments and simulation demonstrate that the **isochoric** lamellar GOA structures achieve minimised resistance for laminar flow when following the corresponding expression of Universal Murray’s law.

(Page 11, Line 46) As shown in Fig. 3d-e and Supplementary Fig. 11a-b, **with the same volume**, the pipe obeying Murray’s law ($\sum r_1^3 = \sum r_2^3 = \sum r_3^3$) achieve minimal resistance for laminar fluid flow both in experiments and in simulation of the scaled-down models (Supplementary Fig. 12).

In response to the last comment, considering the length of the main text, the detailed

theoretical discussion is in Supporting Information: Murray's law in hierarchically tubular network with arbitrary shape. Briefly, we only need a proper variable x showing the power law relationship with sectional area A and transfer rate Q . In this section, we use hydraulic diameter D_H to derive the cubic expression for laminar flow. $D_H = \frac{4A}{P}$, where A is the sectional area and P is the perimeter. Any close geometric shape obviously has a D_H . Other common length variables x , such as side length, diameter, or perimeter, are proportional to the hydraulic diameter D_H . In this way, we get the cubic expression in the main text. The corresponding discussion in Supporting Information is repeated below:

(Supporting Information, Page 13) Considering the laminar flow in a tubular branching network with an unidentified close geometric shape, as shown in Fig. 1c, the hydraulic diameter of the tube is: $D_H = \frac{4A}{P}$, where A is the cross-sectional area and P represents the perimeter.

(Supporting Information, Page 14) Common length variables, such as side length, diameter, or perimeter, are proportional to the hydraulic diameter D_H . Thus, for a common length variable x , $A \propto x^2$, $Q \propto x^4$, and Murray's law in tubular network with arbitrary shape can also be written as the cubic form: $\sum x_1^3 = \sum x_2^3 = \dots = \sum x_i^3$.

We thank all the reviewers again for their comments on our manuscript.

REVIEWERS' COMMENTS

Reviewer #1 (Remarks to the Author):

In this revised version, the authors addressed most of the points raised by the referees in a reasonable way. Overall, I think that the work is valuable and can be accepted now.

Reviewer #2 (Remarks to the Author):

It is a nice research work. The authors have well addressed the questions proposed by the reviewers. I have only one minor concern. The authors mentioned in the Conclusion: "For diffusion, ionic transport, and electronic transfer in arbitrary tubular networks, the optimised law can be transformed into an equation of the sum of pores' cross-sectional area." Considering that Murray's law is important to applications involving mass transfer, for electrons transfer in arbitrary tubular networks, it means the electrons movement or charged-ions movement? About this point, the authors are encouraged to expand the description about the significance of research in the conclusion part. This provides more inspiration for readers.

Reviewer #3 (Remarks to the Author):

The revised manuscript remains largely unchanged from its original form, leaving my initial concerns unaddressed. Given the rigorous standards of Nature Communications, the submission falls short in a few critical areas:

The contribution made in manuscript is entirely theoretical. The experiments only cover laminar flow, which is only one (extremely well-documented) class of problems for a proposed theory that claims to be generic to many physical systems. Moreover, the use of GOA nanostructures does not align with the main topic of the paper. That choice is only restrictive: since you only consider laminar flow in your experiment, it would be natural, and much more versatile, to use a macroscopic geometry, such as a 3D printed structure or just a macroscopic pipe network, which would also allow for turbulent flow or the use of non-Newtonian fluids. That would allow for a wider range of systems to be investigated.

I believe that this manuscript logically consists of two weakly related stories: one theoretical on Murray's law and one experimental on GOA nanostructures. I suggest separating these topics for clarity and targeting suitable venues for each. Consequently, I recommend rejection of this manuscript.

Reviewer #3 (Remarks on code availability):

n/a

RESPONSE TO REVIEWERS' COMMENTS

We sincerely appreciate the reviewers' constructive and overall positive feedback on our revised manuscript. Accordingly, we have carefully reviewed and revised our paper to address the reviewers' remaining comments. Their original comments are marked in **BLUE** here. The changes we have made are marked in **RED** in the revised manuscript and this response.

Reviewer #1

In this revised version, the authors addressed most of the points raised by the referees in a reasonable way. Overall, I think that the work is valuable and can be accepted now.

Reply: Thank you very much for the high evaluation on our manuscript and the acknowledgement of the revisions. Your review has greatly contributed to refining our work.

Reviewer #2

It is a nice research work. The authors have well addressed the questions proposed by the reviewers. I have only one minor concern. The authors mentioned in the Conclusion: "For diffusion, ionic transport, and electronic transfer in arbitrary tubular networks, the optimised law can be transformed into an equation of the sum of pores' cross-sectional area." Considering that Murray's law is important to applications involving mass transfer, for electrons transfer in arbitrary tubular networks, it means the electrons movement or charged-ions movement? About this point, the authors are encouraged to expand the description about the significance of research in the conclusion part. This provides more inspiration for readers.

Reply: Thank you sincerely for your review and affirmative feedback. Your current and previous constructive suggestions have been very crucial in improving our research.

In response to your insightful and important concern here, in short, **Murray's law works in both**. The terms "**ionic transport**" and "**electronic transfer**" follow the prior research on Murray's law and Murray materials (*Nat. Comm.*, 8, 14921; *Small*, 14, 1802670.). These two terms encompass the carrier movement (electrons, holes, or ions) under an electric field in the conductor, semiconductor, or electrolyte, following Ohm's law and Pouillet's law. "Ionic transport" describes the ion's movement in the electrolyte under an electric field, known as ionic migration, focused by the researchers in battery electrodes and supercapacitors. On the other hand, "electronic transfer" describes the current in the conductor or semiconductor.

Accordingly, we replaced the above two terms with "**ionic migration**" and "**electron transportation**" in our manuscript and supplementary information, to keep the correctness in terminology and highlight these two types of mass transfer process. For example, the sentence in Conclusion has now been changed to:

(Page 10, Line 27) For diffusion, **ionic migration** and **electron transportation** in

arbitrary branching networks, the optimised law can be transformed into an equation of the sum of corresponding cross-sectional area.

Additionally, we slightly modified the sentence when we first mentioned these transfer types:

(Page 2, Line 40) Beyond laminar fluid flow, Murray's law takes on a square form for diffusion, **ionic migration in electrolyte under an applied electric field, and electron transportation in conductors or semiconductors** [24]:

Reviewer #3

The revised manuscript remains largely unchanged from its original form, leaving my initial concerns unaddressed. Given the rigorous standards of Nature Communications, the submission falls short in a few critical areas:

The contribution made in manuscript is entirely theoretical. The experiments only cover laminar flow, which is only one (extremely well-documented) class of problems for a proposed theory that claims to be generic to many physical systems. Moreover, the use of GOA nanostructures does not align with the main topic of the paper. That choice is only restrictive: since you only consider laminar flow in your experiment, it would be natural, and much more versatile, to use a macroscopic geometry, such as a 3D printed structure or just a macroscopic pipe network, which would also allow for turbulent flow or the use of non-Newtonian fluids. That would allow for a wider range of systems to be investigated.

I believe that this manuscript logically consists of two weakly related stories: one theoretical on Murray's law and one experimental on GOA nanostructures. I suggest separating these topics for clarity and targeting suitable venues for each. Consequently, I recommend rejection of this manuscript.

Reply: Thank you for the review and the constructive critique of this work. In our last response, we have supplied sufficient experiments and made modifications to address your concerns.

For your concern about the significance and the experiments in laminar flow, we have discussed them in the last response. Laminar flow is the most apparent transport process in living organisms (*Physiol. J.*, 124, 631; *Nature*, 421, 939-942.). Thus, it is the first and most well-documented transfer type for Murray's law. Laminar flow is also crucial in industrial production and microfluidics, where the corresponding Murray's law has been applied (*Chem. Eng. Process.*, 194, 109564.; *Lab on a Chip*, 6, 447-454; *Microfluid. Nanofluidics*, 19, 737-749.). Therefore, we experimented with laminar flow. Our results on laminar flow sufficiently verify the proposed Universal Murray's law and first demonstrate the transfer superiority of obeying Murray's law in porous materials.

In response to your concern about the limitation of verification in laminar flow, we have added dimensionless diffusion simulation to further examine Murray's law and Universal Murray's law. As shown in Supplementary Fig. 19 below, these results have verified the

law with another kind of transfer type besides laminar flow.

Supplementary Figure 19. Dimensionless diffusion simulation of hierarchical structures. (a) Simulated relative diffusion efficiency of scaled-down hierarchical planar structures. (b) Simulated relative diffusion efficiency of scaled-down hierarchical tubular structures. (c) Dimensionless diffusion simulation of scaled-down hierarchical planar structures. (d) Dimensionless diffusion simulation of scaled-down hierarchical tubular structures. The dimensionless Fickian diffusion is simulated with an inlet concentration of 1 and an outlet of 0. A unity diffusion coefficient is applied. The solutions are seen as converged when the residuals reach 10^{-8} . Using the same boundary conditions, different structures' diffusion fluxes are computed and compared with those following Murray's law as relative diffusion efficiency.

Accordingly, we added the following discussion in the main text:

(Page 9, Line 24) Besides laminar flow, the planar and tubular structures following the corresponding Universal Murray's law for diffusion also exhibit optimum diffusion efficiency, as shown in the simulation results in Supplementary Fig. 19.

To further clarify the enquiries about the verification of laminar flow, we added the discussion in the main text:

(Page 8, Line 4) Laminar flow is the most obvious transport process in living organisms [23, 30], and thus, it is the first and most widely-discussed transfer type for Murray's law.

In response to the comment about the use of GOA, this part of the work is tightly closed to the theme and is indispensable for the paper. Firstly, as stated in our title, the topic of this work is to optimise the mass transfer in “**synthetic structure**”. Consequently, we validated and demonstrated the theory **in synthetic porous materials** rather than macroscopic networks. We selected the GOA due to its high porosity, robust mechanical strength, aligned pore geometry, and proper and controllable pore size. Secondly, the reported experiments in the macroscopic network (*J. Fluid Mech.*, 967, A6; *Soft Matter*, 6, 739-742; *Mater. Des.*, 190, 108572.) can hardly adequately represent porous materials with far more pores. It can be calculated that the Murray tubular GOA we used has approximately 16,000 to 260,000 channels in different hierarchies. Techniques like 3D printing are unlikely to construct the macroscopic channels on these numbers and conduct experiments. Based on the above two points, our results on GOA are more persuasive and appealing for the most readership, material scientists.

We thank all the reviewers again for their feedback on our manuscript.